# Trafficked Malayan pangolins contain viral pathogens of humans

Wenqiang Shi[1,9], Mang Shi [2,9], Teng-Cheng Que[3,9], Xiao-Ming Cui[1,4,9], Run-Ze Ye [5,9], Luo-Yuan Xia[1,5], Xin Hou[2], Jia-Jing Zheng[1,6], Na Jia[1,4], Xing Xie[7], Wei-Chen Wu[2], Mei-Hong He[3], Hui-Feng Wang[8], Yong-Jie Wei[3], Ai-Qiong Wu[3], Sheng-Feng Zhang[7], Yu-Sheng Pan[1], Pan-Yu Chen[3], Qian Wang[1,5], Shou-Sheng Li[3], Yan-Li Zhong[3], Ying-Jiao Li[3], Luo-Hao Tan[3], Lin Zhao[5], Jia-Fu Jiang [1,4 ✉], Yan-Ling Hu [7,8 ✉] and Wu-Chun Cao [1,4,5 ✉]

**Pangolins are the most trafficked wild animal in the world according to the World Wildlife Fund. The discovery of SARS-CoV-2-related coronaviruses in Malayan pangolins has piqued interest in the viromes of these wild, scaly-skinned mammals. We sequenced the viromes of 161 pangolins that were smuggled into China and assembled 28 vertebrate-associated viruses, 21 of which have not been previously reported in vertebrates. We named 16 members of *Hunnivirus*, *Pestivirus* and *Copiparvovirus* pangolin-associated viruses. We report that the L-protein has been lost from all hunniviruses identified in pangolins. Sequences of four human-associated viruses were detected in pangolin viromes, including respiratory syncytial virus, *Orthopneumovirus*, *Rotavirus A* and *Mammalian orthoreovirus*. The genomic sequences of five mammal-associated and three tick-associated viruses were also present. Notably, a coronavirus related to HKU4-CoV, which was originally found in bats, was identified. The presence of these viruses in smuggled pangolins identifies these mammals as a potential source of emergent pathogenic viruses.**

Pangolins (Class Mammalia, Order Pholidota) have drawn increasing global attention in terms of their public health importance since the discovery of SARS-CoV-2-related coronaviruses in Malayan pangolins (*Manis javanica*)[1–4]. Consumer demand for pangolin meat and scales has led to pangolins becoming the world's most trafficked wild mammal[5]. More than one million pangolins have been poached in the past ten years, with an estimated 195,000 pangolins traded in 2019 for their scales alone, according to the World Wildlife Fund[6]. Global trade of pangolins undoubtedly increases the risks of transmission of any viruses that they harbour. Although some studies have detected several viruses in a few smuggled, or rescued, pangolins, the general potential risk of viral zoonoses from pangolins remains unclear[7–12]. Furthermore, the interferon epsilon (*IFNE*) gene, which is part of the innate immune defence in most placental mammals, is pseudogenized in pangolins, suggesting that resistance of pangolins to infection may be reduced[13].

To evaluate the potential role of pangolins in the emergence of viral pathogens of humans and animals, we carried out a comprehensive investigation of the viromes of 161 pangolins smuggled into China from Southeast Asia.

## Results

**Characteristics of pangolin samples.** We previously identified SARS-CoV-2-related coronaviruses from Malayan pangolins (*Manis*

*javanica*)[1]. Here we expanded our analysis to a set of 161 pangolins smuggled from Southeast Asia into China in 2018–2019, that were confiscated by Customs officials. These pangolins were most probably trafficked for consumption as 'game meat'. We used previously developed specific reverse transcription polymerase chain reaction (RT-PCR) primers to test for SARS-CoV-2-related coronaviruses[1], and found that all 161 pangolins were negative for SARS-CoV-2 viruses. We managed to collect all the leftovers of archived tissue samples. Only limited amount of muscle, lung, intestine, spleen, liver, heart and kidney tissues were available from different pangolins, hence we pooled the tissues from each individual pangolin into a single sample. Next, we prepared libraries for all 161 pangolins and sequenced the meta-transcriptomes (see Supplementary Table 1 for detailed information about each library). Unfortunately, only archived tissue samples were available, so it was not possible for us to identify the pangolin species through traditional morphological classification. We therefore identified pangolin species on the basis of mitochondrial contigs present in the meta-transcriptome data, and found that all 161 pangolins were *Manis javanica*.

**Diversity in Malayan pangolin viromes.** We generated $1.9 \times 10^{10}$ meta-transcriptome reads, which were assembled and annotated for virus identification and characterization. After quality control, we identified 12.7 M viral reads, representing 0.07% of the total reads, through DIAMOND blastx[14] comparison against the NCBI

¹State Key Laboratory of Pathogen and Biosecurity, Beijing Institute of Microbiology and Epidemiology, Beijing, P. R. China. ²School of Medicine, Shenzhen Campus of Sun Yat-sen University, Sun Yat-sen University, Shenzhen, Guangdong, P. R. China. ³Terrestrial Wildlife Rescue and Epidemic Diseases Surveillance Center of Guangxi, Nanning, Guangxi, P. R. China. ⁴Research Unit of Discovery and Tracing of Natural Focus Diseases, Chinese Academy of Medical Sciences, Beijing, P. R. China. ⁵Institute of EcoHealth, School of Public Health, Cheeloo College of Medicine, Shandong University, Jinan, Shandong, P. R. China. ⁶College of Life Science and Technology, Beijing University of Chemical Technology, Beijing, P. R. China. ⁷Life Sciences Institute, Guangxi Medical University, Nanning, Guangxi, P. R. China. ⁸Center for Genomic and Personalized Medicine, Guangxi Key Laboratory for Genomic and Personalized Medicine, Guangxi Collaborative Innovation Center for Genomic and Personalized Medicine, Guangxi Medical University, Nanning, Guangxi, P. R. China. ⁹These authors contributed equally: Wenqiang Shi, Mang Shi, Teng-Cheng Que, Xiao-Ming Cui, Run-Ze Ye. ✉e-mail: jiangjf2008@gmail.com; huyanling@gxmu.edu.cn; caowuchun@126.com

non-redundant database. Viral reads were subsequently classified into 42 families, each of which was highly variable in terms of prevalence and abundance (Fig. 1a). Vertebrate- and bacteria-associated virus families had higher abundance and prevalence than other families.

We performed phylogenetic analyses on the basis of the amino acid (aa) of the most conserved RNA-dependent RNA polymerase (RdRp) protein for RNA viruses, and capsid protein for DNA viruses. Since some of the newly identified viruses could not be easily categorized according to the current scheme of virus classification[15], we used a previously adopted tactic and incorporated currently defined virus orders, families and floating genera into 'superclades' (Fig. 1b)[16]. We identified sequences of three unclassified viruses, which fell outside known vertebrate-associated viral families and were distinct from well-defined viral families in *Bunyavirales-Arenaviridae* and *Picornavirales-Caliciviridae* superclades, showing only 28.1%–67.6% aa identities of RdRp with the most closely related viruses (Fig. 1b). Despite being present in pangolin libraries, virus sequences that are probably associated with diet or gut microbiome, including bacteriophages, as well as eukaryote-related viruses specific to fungi and plants were excluded and are not discussed further. This left 28 distinct vertebrate-associated viruses that are closely related to virus families or genera well-known to infect vertebrates, with 21 viruses being newly identified in this study (Supplementary Table 2). We then designed specific primers according to assembled virus sequences (Extended Data Fig. 1) and performed RT-PCR (Extended Data Fig. 2) to confirm the presence of viruses identified using the MGISEQ-2000 sequencing platform (Supplementary Table 1), followed by Sanger sequencing (Supplementary Fig. 1). We obtained 94 complete or nearly complete viral genomes of 24 virus species in total.

**Previously unidentified pangolin-associated viruses.** The sequences of 16 viruses in the *Hunnivirus*, *Pestivirus* and *Copiparvovirus* genera formed separate clusters in the phylogenetic tree of each genus and were distinct from other known vertebrate viruses reported previously (Fig. 2), suggesting that they might have circulated within pangolins for extended periods of time. We provisionally named them pangolin-associated viruses.

Five pangolin-associated viruses (named *Pangolin hunnivirus BIME 1–5*) had only 62.3%–69.4% aa identity of RdRp with the closest species (Fig. 2a), which was isolated from a rat in the United States (GenBank accession no. NC_025675.1)[18] and was from the *Hunnivirus* genus in the *Picornaviridae* family. Many hunniviruses cause mucocutaneous, encephalic, cardiac, hepatic, neurological or respiratory diseases in a wide range of vertebrates[17]. Phylogenetic analyses based on the polyprotein revealed that the 10 hunniviral sequences from the 5 viruses identified in this study formed a cluster distinct from previously known *Hunnivirus* isolates (Fig. 2a and Extended Data Fig. 3a).

Seven whole genome sequences of the viruses (except for one *Pangolin hunnivirus BIME 2*, one *Pangolin hunnivirus BIME 3* and one *Pangolin hunnivirus BIME 4*, which were incomplete due to a relatively low abundance of viral reads) were obtained, these sequences sharing 73.0%–77.0% nucleotide (nt) identities with each other and only 60.5%–61.7% nt identities with the most closely related sequence from the US rat[18]. A phylogenetic tree based on the whole genome sequence as well as sequences of various proteins had topological structures similar to that based on the polyprotein (Extended Data Fig. 4), which confirmed that these seven genomes represented new members of the *Hunnivirus* genus.

Next, we analysed genomic structures and found that all seven hunniviral genomes from pangolins had lost the L-protein (Fig. 2b). The L-protein is known to play an important role in interferon antagonism during early viral infection according to an in vivo experiment[19]. Considering the pangolin as an *IFNE*-deficient

animal[13], loss of L-protein suggests that these viruses might have adapted to pangolin hosts. In addition, a series of three deletions with a total of 82–85 aa were detected in 2B protein (Fig. 2c), which has known functions in membrane permeability, cell death and host immune responses[20]. The large fragment deletion in 2B protein will most probably influence these functions and subsequently change viral infectivity or pathogenicity through viral evasion of the host immune response.

Sequences of nine distinct virus species were newly identified in the genus *Pestivirus* of family *Flaviviridae*, members of which are known to cause asymptomatic infection or produce a range of clinical conditions such as acute diarrhoea, acute haemorrhagic syndrome, acute fatal disease, and a wasting disease in pigs and ruminants[21,22]. In the phylogenetic tree (Fig. 2d and Extended Data Fig. 3b), the newly identified pangolin pestiviruses (*Pangolin pestivirus BIME 1–9*) were clustered with *Dongyang pangolin virus*[7], although they only had 69.6%–88.8% aa identities with each other in RdRp. Whole genomes of the nine viruses were assembled from 22 pangolin samples, indicating richness of these viruses in pangolins (Extended Data Fig. 5a). The high prevalence and abundance of pestivirus genomes in our samples may suggest that pangolins are a favoured host. Indeed, *Dongyang pangolin virus* was previously detected in pangolins that died from a haemorrhagic disease[7], suggesting that the pestiviruses we detected might be pathogenic. Furthermore, we observed that the genome identity of the pangolin pestiviruses varied greatly across their 11 encoded proteins (Supplementary Table 3). Capsid proteins had the highest aa identity (64.9%–76.6%), while p7 proteins exhibited the lowest degree of conservation (20.0%–30.0%) compared with porcine pestivirus from Australia (GenBank accession no. NC_023176.1).

We detected sequences of two viruses in the *Copiparvovirus* genus (*Parvoviridae* family), which contains diverse viruses capable of infecting a wide range of vertebrates[23]. A phylogenetic tree based on capsid protein showed that the two viruses detected in eight pangolins were clustered in a separate clade in the *Copiparvovirus* genus (Fig. 2e and Extended Data Fig. 3c) and were most closely related to a horse parvovirus isolated in the United States (GenBank accession no. KR902500.1). We assembled six full genomes that shared 80.4%–81.4% identities with each other and 49.5%–50.4% identities with genomes of viruses previously reported in this genus (Extended Data Fig. 5b).

**Human-associated viruses detected in pangolins.** Sequences of four viruses associated with human infections were detected in pangolins. Human respiratory syncytial virus subtype A (RSV-A) within the genus *Orthopneumovirus* of family *Pneumoviridae*—the common pathogen that causes severe respiratory disease in children under two years of age and older people[24,25]—was found in two pangolin samples. Phylogenetic analysis revealed that the two RSV-As from pangolins clustered with human RSV-As identified around the world and with pangolin RSV-As detected in another batch of pangolin samples[26] (Fig. 3a and Extended Data Fig. 3d). The whole genome sequences of the two RSV-As had 100% nt identity with each other, 99.98%–100% identity with previously reported pangolin RSV-As and 99.40%–99.64% identity with their closest human RSV-A from Australia in 2017[27]. RSV was first identified in chimpanzees and later documented to be a mainly human pathogen[28]. The identical RSV-A sequences suggest that the two pangolins might be infected from the same infected person or transmitted virus from each other. In addition to RSV-As, we also found a new species of the genus *Orthopneumovirus* (*Orthopneumovirus BIME 1*), which formed a lineage separated from the RSV-A clade and shared only a 65.6% aa identity with RSV-As in RdRp. The newly identified pangolin orthopneumovirus related to human RSVs raises an important question for future research: are pangolins intermediate or reservoir hosts of human RSVs?

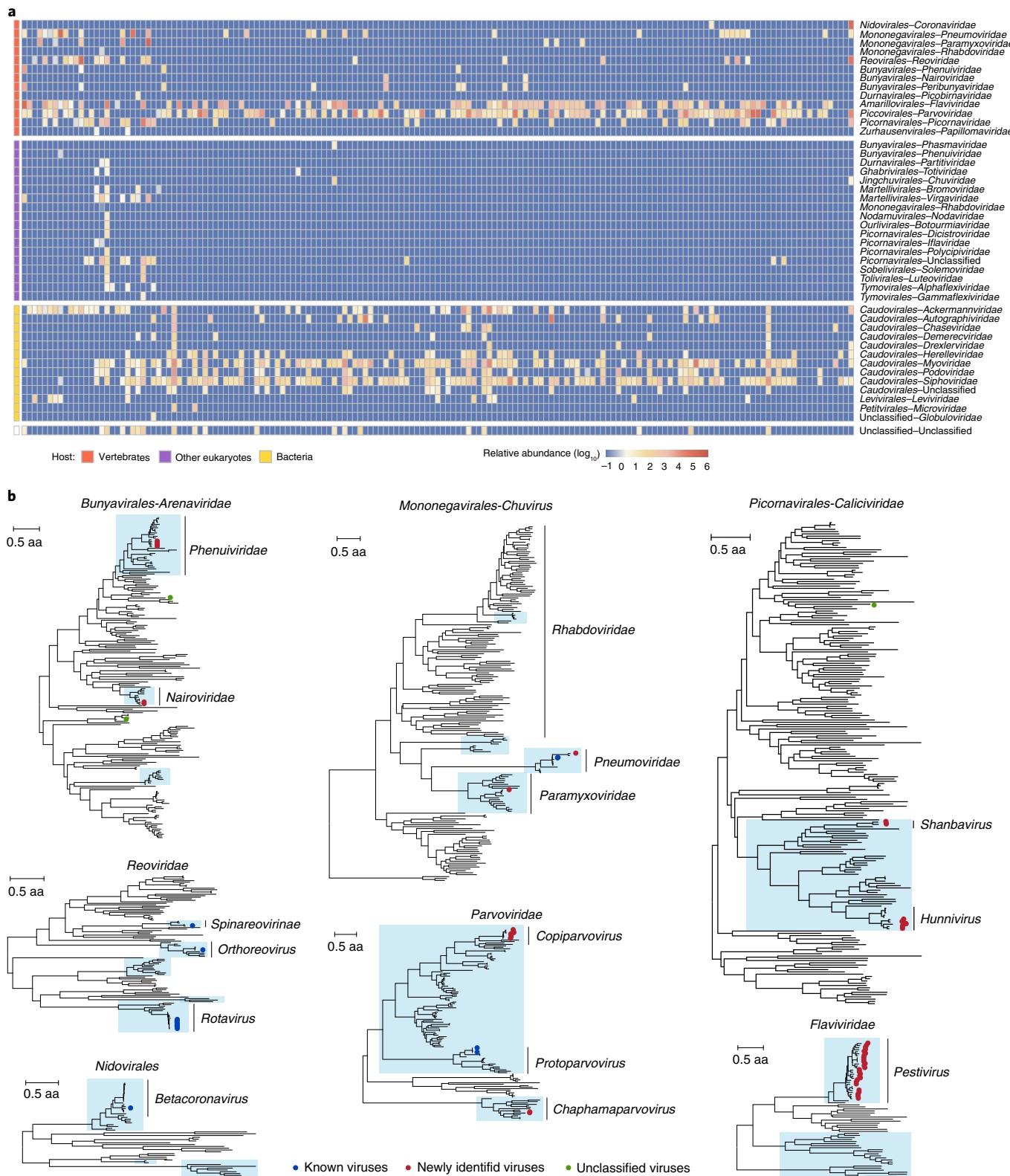

**Fig. 1 | Diverse viromes of pangolins. a**, Virus abundance profile across pangolin samples. Each cell in the heat map represents the normalized number of reads belonging to the given virus order and family according to blastx comparison. Viruses were grouped according to host category, including vertebrates, other eukaryotes, bacteria and unclassified. For viral families with multiple host categories, the host category of each viral read was inferred on the basis of the host of the genus assigned by blastx with $E$-value $<1 \times 10^{-5}$. **b**, Phylogenetic trees were constructed on the basis of RdRp protein amino acid for RNA viruses and capsid protein for DNA viruses. Viruses discovered in this study are labelled with solid circles: red, newly identified viruses; blue, known viruses; green, unclassified viruses. Vertebrate-associated viruses are shaded in light blue. Families without assembled contigs containing the RdRp domain were not displayed in Fig. 1b, although the viral reads were detected in Fig. 1a.

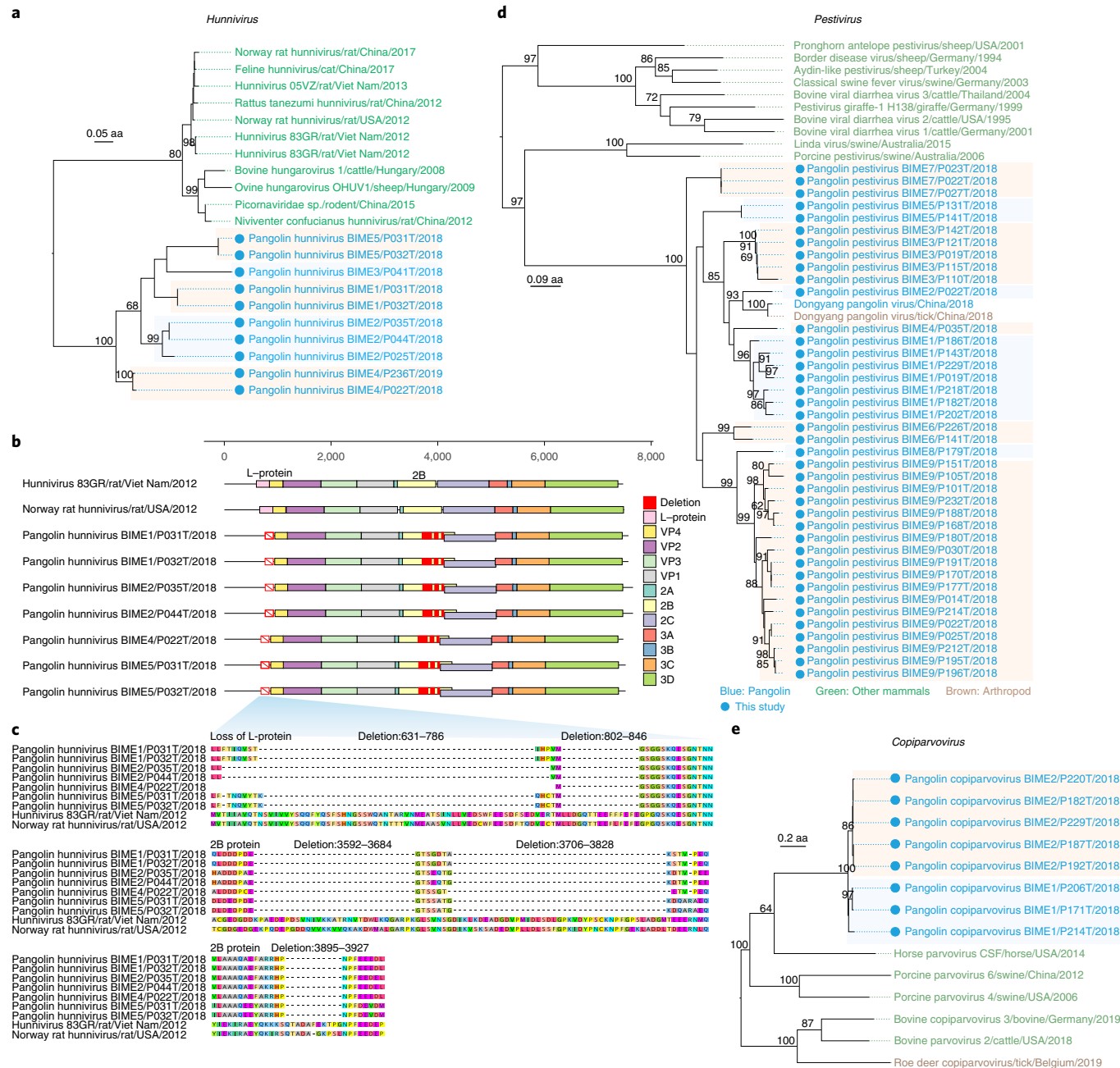

**Fig. 2 | Analyses of pangolin-associated virus sequences. a**, Phylogeny of viruses in the genus *Hunnivirus* based on the amino acid of polyprotein. **b**, Genome organization of *Pangolin hunnivirus* in comparison with two closest related species. The loss of the L-protein gene is indicated by a red rectangle with diagonal line, and large sequence deletion is marked by a red box. **c**, Sequence deletion in L-protein and 2B protein of *Pangolin hunnivirus*. **d**, Phylogeny of viruses in the genus *Pestivirus* based on polyprotein. **e**, Phylogeny of viruses in the genus *Copiparvovirus* based on capsid protein. Tree tips are coloured according to host type: blue, pangolins; green, other mammals; brown, arthropods. Viruses identified in this study are marked by solid blue circles. The tree is mid-point rooted for visualization only.

*Rotavirus A* within the genus *Rotavirus* of family *Reoviridae*, which causes diarrhoea in children and animals[29], was detected in seven pangolin samples. Phylogenetic analysis based on RdRp revealed that the sequences of pangolin rotavirus A formed two close clusters (Fig. 3b and Extended Data Fig. 3e). Within each cluster, the sequences were nearly identical, suggesting that the viruses might have come from the same pangolin cage or the same wet market during the smuggling and illegal trade. Of the 11 genomic segments, assembled viral sequences of the same segment showed 93.37%–100% nt identities and 52.0%–87.7% nt identities with the known strains. Notably, the sequences of five segments (that is,

VP2, VP4, NSP1, VP6 and NSP5/NSP6) in pangolin rotavirus A were distinct from established genotypes and were classified as new genotypes according to the criteria recommended by the Rotavirus Classification Working Group[30] (Supplementary Table 4). These findings suggest either that *Rotavirus A* may have associated with pangolin for an extensive period of time, or that we failed to cover the diversity gap between pangolin rotaviruses and those identified from other mammalian hosts.

In addition, the sequence of a *Mammalian orthoreovirus* within the genus *Orthoreovirus* of family *Reoviridae* was also obtained from a pangolin sample. Phylogenetic analysis based on RdRp showed

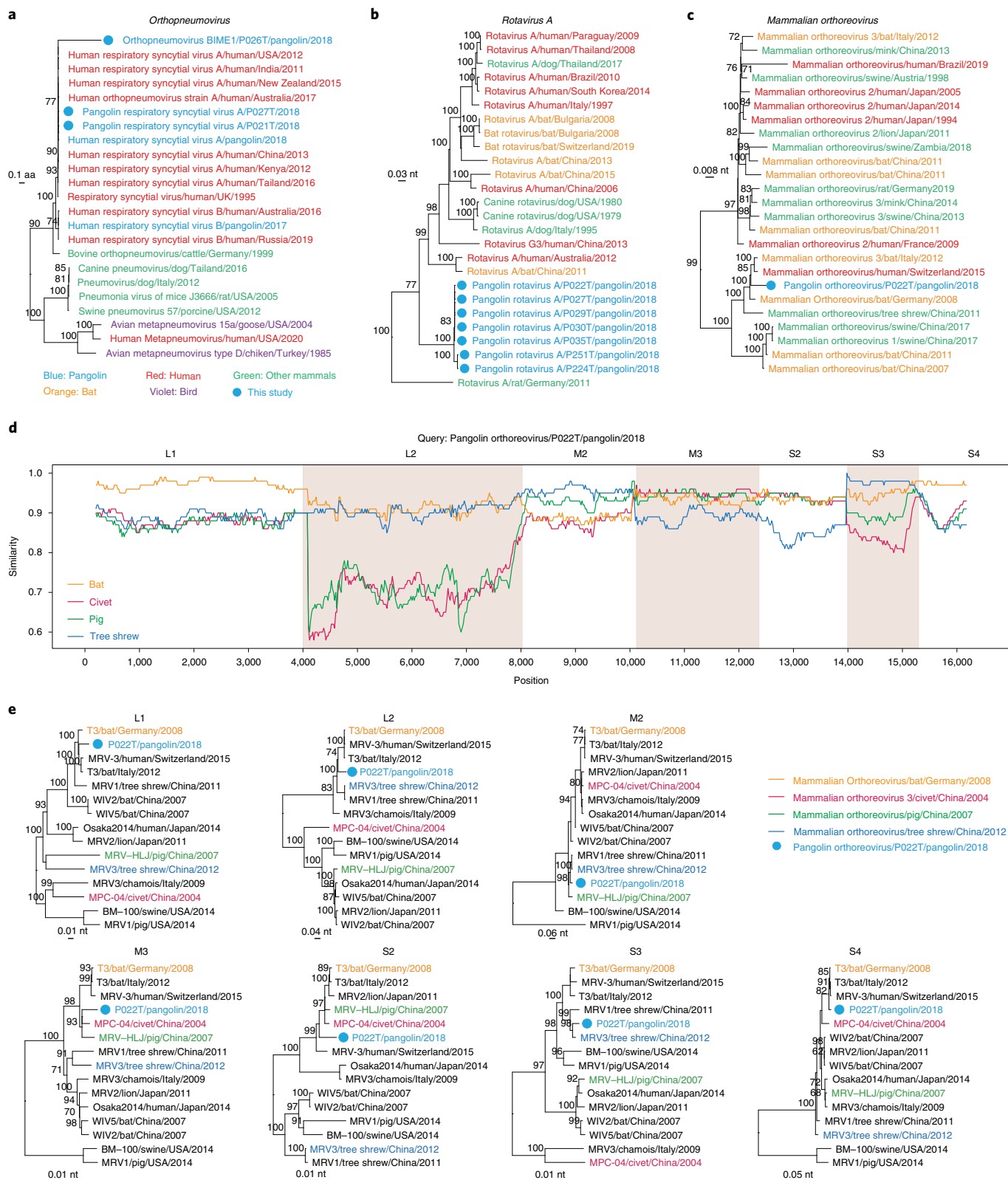

**Fig. 3 | Analyses of human-associated virus sequences. a**, Phylogeny of viruses in the genus *Orthopneumovirus* based on RdRp protein. **b**, Phylogeny of *Rotavirus A* based on the VP1 (that is, RdRp) gene. **c**, Phylogeny of *Mammalian orthoreovirus* based on the RdRp gene. Tree tips are coloured according to host type: red, humans; blue, pangolins; orange, bats; green, other mammals; purple, birds. Viruses newly identified in this study are marked by solid blue circles. **d**, Sliding window analysis of changing patterns of sequence similarity using *Pangolin orthoreovirus* against strains from marked palm civet, pig, bat and tree shrew. In the scanning, the sliding window size was set to 400 bp and the step size was 20 bp. **e**, Phylogenetic trees of different segments of *Mammalian orthoreovirus*. Branch supports obtained from 1,000 bootstrap replicates are shown. The *Pangolin orthoreovirus* is indicated in blue. The strains from marked palm civet, pig, bat and tree shrew are indicated in red, green, yellow and dark blue, respectively.

that the *Pangolin orthoreovirus* clustered with a bat (*Plecotus auritus*) strain from Germany with 97.7% nt identity (GenBank accession no. JQ412755.1)[31], which was phylogenetically close to the *Mammalian orthoreovirus* from a human in Switzerland (Fig. 3c and Extended Data Fig. 3f). Interestingly, we observed the signals of reassortment between the pangolin orthoreovirus strain and other strains from a masked palm civet (*Paguma larvata*)[32], a pig in China[33], a bat (*Plecotus auritus*) in Germany[31] and a tree shrew (*Tupaia belangeri*) in China (GenBank accession no. MG451071.1) (Fig. 3d), suggesting cross-species transmission and co-infection of orthoreoviruses among multiple animal hosts. Phylogenetic analyses revealed discrepancies in the clustering of the seven segments between *Pangolin orthoreovirus* and the strains from other hosts. *Pangolin orthoreovirus* clustered with the bat strain in the L1, L2 and S4 segments, with the civet and pig strains in the M3 and S2 segments, and with the tree shrew strain in the M2 and S3 segments (Fig. 3e). Reassortment has been frequently observed in segmented virus and is considered as a primary mechanism for interspecies transmission and the emergence of novel strains[34]. A cross-family recombinant from orthoreovirus and coronavirus has been reported in bats[35], suggesting that distinct recombination events might occur in orthoreovirus and increase the potential for spillover.

**Animal-associated viruses detected in pangolins.** We obtained the sequence of an HKU4 coronavirus strain P251T, which had 96.7% aa identity in RdRp with *Tylonycteris*-bat-CoV-HKU4 from lesser bamboo bats[36]. Pangolin-CoV-HKU4-P251T was located at the root of bat HKU4 coronavirus clade (Fig. 4a and Extended Data Fig. 3g). Considering that *Tylonycteris*-bat-CoV-HKU4 possesses a spike (S) protein capable of utilizing the MERS-CoV receptor human dipeptidyl-peptidase-4 (hDPP4)[37], we assessed the receptor binding domain (RBD) similarity of MERS-CoV-related coronaviruses. Both pangolin-CoV-HKU4-P251T and *Tylonycteris*-bat-CoV-HKU4 shared 4 of 10 key residues in the RBD of MERS-CoV[38] (Extended Data Fig. 6a). Although pangolin-CoV-HKU4-P251T was distantly related to MERS-CoV for most of the genome, their relationship was much closer in the RBD region (Extended Data Fig. 6b). In terms of hosts, pangolin DPP4 showed higher aa similarity with human DPP4 (89.0%–89.3%) than bat DPP4 (82.4%–83.2%), suggesting a risk for a more probable spillover event of pangolin-CoV-HKU4-P251T through the utilization of human DPP4. Another bat-associated virus, a novel species of the genus *Shanbavirus* of family *Piconaviridae*, was first recognized in pangolins. Phylogenetic analysis showed that it was most closely related to *Shanbavirus A* obtained from bent-winged bat (*Miniopterus fuliginosus*) in China[39], with only 62.2% aa identity in RdRp (Fig. 4b and Extended Data Fig. 3h).

Besides bat-associated virus sequences, we obtained the sequences of other mammal-associated viruses including rodent- and canine-associated viruses. Four sequences of a virus in the genus *Respirovirus* of family *Paramyxoviridae* were detected, these sequences clustering in the same lineage as a previously reported pangolin respirovirus and *Sendai virus* circulating in mice from Asia (Fig. 4c and Extended Data Fig. 3i). Its genome shared high similarity with that of a respirovirus previously reported in pangolin (93.1%) and *Sendai virus* (89.2%)[8]. Another rodent-associated virus was a new species (*Pangolin chaphamaparvovirus BIME 1*) in the genus *Chaphamaparvovirus* of family *Parvoviridae*. It had only 65.9% aa identity of capsid protein with the closest *Rat parvovirus 2* from a rat in China[40], although they were in the same lineage (Fig. 4d and Extended Data Fig. 3j). Lastly, whole genome sequences of pangolin protoparvovirus in the *Protoparvoirus* genus of *Parvoviridae* family were identified in 23 pangolin samples. They were nearly identical to each other and to canine parvovirus from dogs in China (Fig. 4e and Extended Data Fig. 3k)[41], with a genomic identity >99.0%. These sequences of diverse mammal-associated

viruses suggest that multiple cross-species transmission events have occurred between pangolins and other mammals.

We identified sequences of three virus species related to tick-associated viruses in pangolins, although they were detected from a couple of samples. Malayan pangolins were previously reported to be heavily infested with ticks and subsequently became infected by tick-borne agents[42]. The genomes of viruses in the genus *Phlebovirus* of family *Phenuiviridae*, which includes well-known tick-borne viruses from around the world[43], were detected in two pangolins. In the phylogenetic tree, the *Pangolin phlebovirus BIME 1* in this study was in the same branch as two similar viruses in ticks from Japan[44] (Fig. 4f and Extended Data Fig. 3l). The L, M and S segments of the two strains of phlebovirus had 98.8%–99.2% nt identity with each other, and 63.2%–74.0% identities with the tick-associated viruses in Japan (GenBank accession no. LC133178.1).

Another virus species in the genus *Orthonairovirus* of family *Nairoviridae* clustered with tick-associated viruses such as *Wenzhou tick virus* and *Songling virus*, which were recently proven to be associated with human febrile illness[45] (Fig. 4g and Extended Data Fig. 3m). The nt identities of the L, M and S segments were 67.5%, 61.1% and 64.4%, respectively, between the *Pangolin orthonairovirus BIME 1* and the closest *Wenzhou tick virus*[46]. We also detected a sequence of *Lishui pangolin virus* in the genus *Coltivirus* of family *Reoviridae*, which was recently reported in a pangolin with fatal disease[7]. In the phylogenetic tree based on RdRp, the virus identified in this study formed a distinct lineage together with previously reported *Lishui pangolin virus* and *Shelly headland virus* detected in ticks (*Ixodes holocyclus*) collected from bandicoot (*Bandicota bengalensis*) in Australia (Fig. 4h and Extended Data Fig. 3n)[47]. The presence of tick-associated virus sequences in pangolins reveals risks for transmission of pangolin viruses to vertebrates via tick bites.

**Viral diversity in pangolins.** Because all the Malayan pangolins were smuggled to China with no information about their original habitats, we carried out phylogeographic analyses based on pangolin mitochondrial variant sequences to infer population groups. After excluding samples with poor quality sequencing data, 129 Malayan pangolins from our study, together with 12 Malayan pangolins with known geographic origins or population groups[48], were included. We found that pangolins could be classified into five distinct population groups (Fig. 5a). Pangolins in groups 1–3 clustered with samples from Southeast Asian islands, while group 4 contained samples from inland Asia (Yunnan Province of China and Myanmar). Group 5, which comprised only three samples, formed a distant lineage of Malayan pangolins with no known origin.

To investigate virus diversity in relation to these population groups of Malayan pangolins, we plotted the presence and normalized abundance of each identified virus in each pangolin (Fig. 5b). A water control with sequencing reagents was included on the same run with the sequenced samples to validate the viral abundance estimation (Supplementary Fig. 2). Overall, population groups 1–4 had many pangolin-associated viruses in the genera *Pestivirus* (46.0%) and *Copiparvovirus* (24.6%), followed by mammal-associated viruses in the genus *Protoparvovirus* (23.0%). Malayan pangolins in group 3 had significantly higher viral diversity than those in group 1 (two-sided Wilcoxon rank-sum test, $P = 0.006$) (Fig. 5c). There was no significant difference in virus diversity between pangolins that were possibly from Southeast Asian islands (including groups 1, 2 and 3) and those possibly from inland Asia (group 4).

## Discussion

We carried out an analysis of viruses that can infect wild pangolins and identified diverse sequences of known and unknown viruses, including pangolin-, human-, mammal- and tick-associated viruses. Currently, Malayan pangolins are designated 'critically endangered' by the International Union for the Conservation of

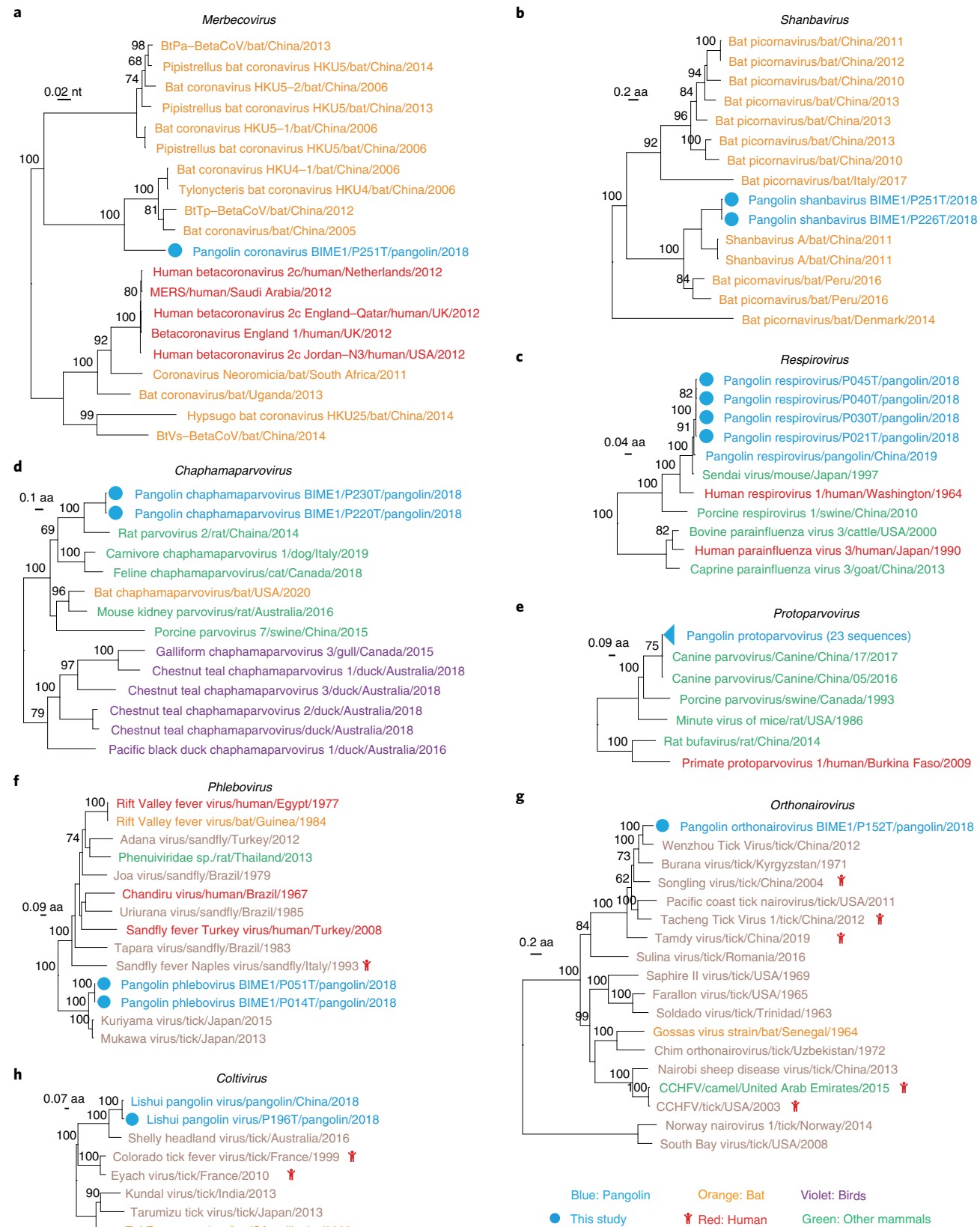

**Fig. 4 | Phylogeny of other mammal-associated viruses in pangolins. a**, Phylogeny of coronaviruses in the subgenus *Merbecovirus* based on the nucleotide sequences of the RdRp domain. **b**, Phylogeny of viruses in the genus *Shanbavirus* based on polyprotein. **c**, Phylogeny of viruses in the genus *Chaphamaparvovirus* based on capsid protein. **d**, Phylogeny of viruses in the genus *Respirovirus* based on RdRp protein. **e**, Phylogeny of viruses in the genus *Protoparvovirus* based on capsid protein. **f**, Phylogeny of viruses in the genus *Phlebovirus* based on RdRp protein. **g**, Phylogeny of viruses in the genus *Orthonairovirus* based on RdRp protein. **h**, Phylogeny of viruses in the genus *Coltivirus* based on RdRp protein. Tree tips are coloured according to host type: red, humans; blue, pangolins; orange, bats; green, other mammals; violet, birds. Viruses newly identified in this study are marked by solid blue circles.

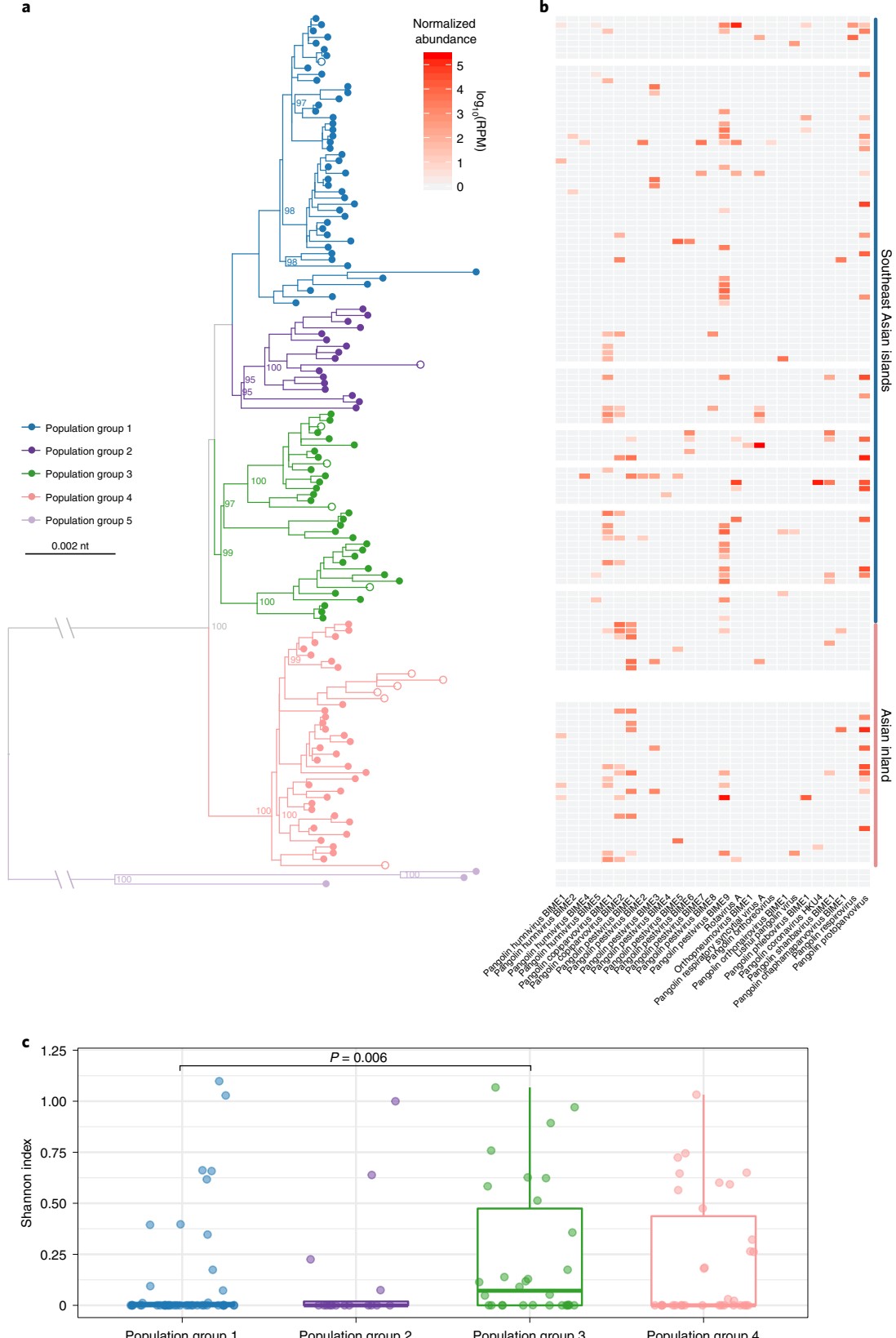

**Fig. 5 | Virus diversity in pangolin population groups. a**, Phylogenetic tree of pangolin population based on mitochondrial variants. Samples collected in this study are indicated by solid circles, and samples with known geographical locations or population groups from public datasets are labelled with open circles. **b**, Viral abundance. Each cell in the heat map represents the normalized number of reads belonging to the given virus and the pangolin sample in the phylogenetic tree. The blank lines in the heat map indicate the samples from public datasets. **c**, Shannon index of viruses in the pangolin population groups 1–4 ($n_{group1}=46$, $n_{group2}=16$, $n_{group3}=30$, $n_{group4}=34$). Boxplot elements: centre line, median; box limits, upper and lower quartiles; whiskers (error bars), the highest and lowest points within 1.5 interquartile range of the upper and lower quartiles. The $P$ value was calculated using a two-sided Wilcoxon rank-sum test.

Nature Red List[49]. We obtained a substantial number of Malayan pangolin samples in this study, made possible only because these endangered animals were trafficked and seized by customs officials. Meta-transcriptome data enabled us to learn more about pangolins as virus hosts, and provided a more complete view of the potential for pangolins to contribute emerging and re-emerging viruses. To better evaluate the viral composition and potential risks to public health from pangolin smuggling, we propose that samples be routinely collected when animals are recovered by customs agents.

Sequences of 16 pangolin-associated virus species belonged to three genera (that is, *Hunnivirus*, *Pestivirus* and *Copiparvovirus*), with high genetic diversity in each genus. These viruses were prevalent in our pangolin cohort. Furthermore, genomes of viruses in the genus *Hunnivirus* identified in these pangolins have all lost the L-protein region, suggesting that these viruses might have evolutionarily adapted to pangolins. We hypothesize that pangolins may be natural reservoir hosts of hunniviruses.

The sequences of the other 12 virus species identified in pangolins are human-associated, mammal-associated and tick-associated, which are phylogenetically related to those of viruses from humans, various animals such as civets, bats, rodents and dogs, as well as ticks. Because *IFNE* that can establish a first line of defence against pathogens in other placental mammals is pseudogenized in pangolins[13], the *IFNE*-deficient animal might be more susceptible to infections and easily infected by various pathogens from other animals or humans, especially under the cramped conditions of illegal trade. For instance, an individual pangolin sample (P251T) contained sequences of *Rotavirus A*, Malayan-CoV-HKU4, *Shannbavirus* and *Protoparvovirus*, further supporting the notion of susceptibility of pangolins to viruses and possible complex exposure networks of pangolins to various animals[50]. The presence of diverse viruses in smuggled pangolins suggests that the conditions of wildlife transportation and sale may facilitate cross-species transmission of these viruses and might result in viral emergence from captured wildlife or wet markets.

One limitation of our study is the pooling of different tissues of a single pangolin into one sequencing library, which means that the distribution of viruses in different organs or tissues cannot be determined. We also recovered unclassified viral contigs that do not belong to any known viral family, and future work will be needed to identify these possible new viruses. Finally, because only archived samples were available from the confiscated pangolins, accurate ecological information and trade routes of smuggled pangolins are unknown, hence we cannot trace where and when pangolins were originally infected with these viruses.

The diverse virus sequences identified in our study imply that pangolins are important in both public and veterinary health. Trading of live pangolins, or products derived from their scales or flesh, will undoubtedly increase the risk of cross-species transmission of viral infections. People need to be made aware of the potential for viruses in pangolins to emerge as human pathogens, and must be prohibited from capturing and eating them.

## Methods

**Sample collection.** The pangolins sampled in this study were intercepted by Guangxi Customs during anti-smuggling operations. Samples were collected between 2018 and 2019. The archived tissue samples, including muscle, lung, intestine, liver, spleen, heart and kidney (Supplementary Table 1), were collected and kept in a −80 °C freezer for further processing.

**RNA extraction, library preparation and sequencing.** Available tissues from the same pangolin were pooled as a single sample and homogenized in PBS solution, and the supernatant was filtered through a 0.45 μM filter column. For the MGISEQ-2000 high-throughput sequencing, total RNA was extracted using a High Pure viral RNA kit (Roche Diagnostics, 11858882001), and viral RNA was then enriched by a Nucleic Acid Microbes purification kit (BGI PathoGenesis Pharmaceutical Technology). Reverse transcription and second-strand synthesis were performed using the PrimeScript double strand cDNA synthesis kit

(Takara Biotechnology, 6111A). A sequencing library was constructed using the QIAGEN QIAseq FX DNA library kit (Qiagen, 180477). The RNA quantity of each constructed library was measured using a Qubit 4.0 fluorometer. The library fragment length was estimated using Qsep-100 (Hangzhou Houze Biotechnology, Qseq100-a). After circularization and generation of DNA nanoballs, paired-end sequencing (2 × 150 bp) of the resulting libraries was performed on the MGISEQ-2000 platform (MGI). To identify and eliminate possible sequencing or reagent contaminants, we included sterile water as well as reagent mix as controls[51]. The controls were sequenced in the same chip with the libraries of pangolin samples. For the Illumina next-generation sequencing, the sequencing library was constructed using NEBNext Ultra II Directional RNA library preparation kit for Illumina. Paired-end (2 × 150 bp) sequencing of the RNA library was performed on an Illumina Novaseq 6000 platform at Annoroad Gene Technology Beijing. Supplementary Table 1 provides detailed information about each library of the 161 pangolin samples.

**Viral contig assembly and annotation.** Adaptor sequences and low-quality bases were removed from raw sequencing reads by the fastp programme (v0.21.0)[52]. The resulting reads were subsequently de novo assembled into contigs using the MEGAHIT programme (v1.2.9)[53] with default parameters. These contigs were then compared to the non-redundant protein database using the DIAMOND blastx programme (v0.9.21)[14] with an $E$-value cut-off of $1 \times 10^{-5}$. To identify viral sequences, taxonomic lineage information was obtained for the top blast hit of each contig, and those annotated under the kingdom 'Viruses' were initially identified as potential virus-associated sequences. To exclude false positives, these potential viral contigs were subjected to blastn comparisons against non-redundant nucleotide databases to distinguish viral sequences from non-viral host sequences, endogenous viral elements and artificial vector sequences. The resulting complete and nearly complete viral genome sequences were further subjected to manual validation by inspecting the mapped reads against the corresponding genomes. The assembled viral genomes were annotated using Geneious (version 2021.2.2)[54]. Species assignment of these confirmed viral genomic sequences was carried out following the International Committee on Taxonomy of Viruses (ICTV) species demarcation criteria of each genus[15]. ICTV species demarcation criteria of each genus are listed in Supplementary Table 5. If ICTV lacks clear criteria in the genera, we used a threshold of amino acid identity of 90% for the viral RdRp (RNA viruses) or conserved replication-associated proteins (DNA viruses)[55,56]. If a detected virus sequence was below the 90% identity threshold, it was designated as a newly identified virus species.

**Quantification of virus abundance.** Two approaches were used to estimate virus abundance at family and species levels. First, quality-controlled reads were compared with the SILIVA database (v138.1, www.arb-silva.de)[57] and pangolin genomes (GCF_014570555.1 and GCF_014570535.1) to filter reads associated with ribosomal RNA and host genomes, respectively, using Bowtie2 (v2.3.4.2)[58]. To estimate virus abundance at family level, the remaining reads were compared to the non-redundant protein database using the DIAMOND blastx programme (v0.9.21)[14] with an $E$-value cut-off of $1 \times 10^{-5}$. Potential viral reads were inferred on the basis of taxonomic lineage information of the top blast hit. To exclude false positives, these potential viral reads were then subjected to blastn comparisons against non-redundant nucleotide databases to exclude non-viral host sequences, endogenous viral elements and artificial vector sequences. The abundance of each virus family was quantified as the number of identified reads per million total filtered reads (RPM) in the library. To estimate the abundance of each new virus species, a read mapping approach was used. The remaining reads after rRNA and host genome filtering were mapped to assembled viral contigs, and then the abundance of each virus was quantified as RPM. To minimize false positives, we applied threshold criteria on the basis of RPM ≥1 and number of identified reads ≥10 for detected viruses. The Shannon index of vertebrate-associated viruses was calculated using the vegan package (v2.5-7)[59].

We compared the presence of viral sequences and their abundance across sequencing libraries. In case identical or near-identical viral sequences were observed in multiple libraries, we used the following steps to rule out possible contamination. We first estimated the read ratio between the highest abundance library and other lower abundance libraries in the same chip. If the ratio was below the index-hopping rate of the sequencing platforms, the reads in the low abundance library were considered as cross-contamination during library preparation and were excluded from further analysis[51]. To eliminate possible index-hopping, we chose the highest index-hopping rate of 0.1% as the threshold[60,61] in the study.

**Confirming the presence of detected viruses by RT–PCR followed by Sanger sequencing.** To further verify the presence of identified virus sequences rather than from contamination, we performed RT-PCR assays of the available samples of the detected viruses in meta-transcriptome sequencing. The specific RT-PCR primers were designed according to assembled virus sequences (Extended Data Fig. 1). The target RT-PCR products were validated by Sanger sequencing (Supplementary Fig. 1). For *Mammalian orthoreovirus*, RT-PCR was also performed to fill gaps in the L1 segment that contains the RdRp gene.

**Phylogenetic analysis.** We first categorized the viral sequences into major viral clades on the basis of the DIAMOND blastx[14] results. Phylogenetic trees were constructed for each of the viral clades on the basis of conserved viral proteins for more accurate taxonomic assignment of the newly identified viruses. For this purpose, sequences were first aligned with related viral sequences within the clade using the programme MAFFT (v7.475)[62]. The ambiguously aligned regions were trimmed using TrimAl (v1.4.rev22)[63] and manually edited. Maximum likelihood trees were subsequently reconstructed on the basis of sequence alignment using the PhyML v3.1 programme[64], employing an LG amino acid substitution model and GTR+gamma nucleotide substitution model with 1,000 bootstrap replicates. To further confirm the topology of the phylogenetic tree of each virus genus, phylogenetic analyses were performed using MrBayes (v3.2.7)[65] (10 million generations) with the same models, using the maximum likelihood method. The MrBayes trees are provided in Extended Data Fig. 3. Sequence similarity analysis was performed using SimPlot (v3.5.1)[66].

**Identification of population groups of pangolins based on mitochondrial sequences.** For each sample, we compared assembled contigs against a database of available pangolin mitochondrial sequences (Supplementary Table 6) using blastn in BLAST software (v2.3.0+)[67] with an *E*-value <$1 \times 10^{-5}$. Pangolin species were inferred on the basis of the top hits. We further included external sequencing data of pangolins (SRR9018586, SRR9018599, SRR9018623, SRR9018628, SRR9018633, SRR9018647, SRR9018652, SRR9018656, SRR9018658, SRR9018665, SRR9018673 and SRR9018674) with known geographical location or population groups[48]. We then called variants on mitochondrial genome using the GATK4 (v4.1.2.0) joint genotype pipeline[68], and individuals with read depth ≥3 and quality ≥30, and individuals with genotype rate >0.9 were included to obtain mitochondrial consensus sequence of each sample using bcftools (v1.9)[69]. Sequences were then aligned using MAFFT (v7.475)[62] and the phylogenetic tree was built using IQTREE (v1.6.10)[70] with the best-fit substitution model and ultrafast bootstrap of 1,000 replicates.

**Reporting summary.** Further information on research design is available in the Nature Research Reporting Summary linked to this article.

## Data availability

The sequencing reads of 161 pangolins after quality control have been deposited in the NCBI Sequence Read Archive (SRA) database under BioProject accession number PRJNA845961. Pangolin mitochondrial sequences and viral sequences assembled in this study were deposited in GenBank and accession numbers are listed in Supplementary Tables 1 and 2. We have provided the GenBank accession numbers of each reference sequence in phylogenetic tree files (newick format) and associated sequence alignments (fasta format) available at https://doi.org/10.6084/m9.figshare.19499030. Source data are provided with this paper.

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

## Acknowledgements

We thank Alibaba Cloud Computing Co., Ltd. for providing computational resources. This work was supported by research grants from the Natural Science Foundation of China (81621005, W.-C.C.; 72071207, W.S.), the State Key Research Development Program of China (2019YFC1200500, J.-F.J.), Guangxi Scientific and Technological Research (GUIKE AB20059002, Y.-L.H.), the Guangxi Medical University Training Program for Distinguished Young Scholars, the Guangxi Key Research and Development Program (Nos. Guike2020AB39264, Y.-L.H.), the Shenzhen Science and Technology Program (KQTD20200820145822023, M.S.) and the Guangdong Province 'Pearl River Talent Plan' Innovation and Entrepreneurship Team Project (2019ZT08Y464, M.S.).

## Author contributions

W.-C.C., J.-F.J. and Y.-L.H. conceptualized the project; W.S., M.S. and R.-Z.Y. developed the methodology; T.-C.Q., A.-Q.W., X.-M.C., M.-H.H., Y.-J.W., S.-S.L., Y.-L.Z., Y.-J.L., P.-Y.C., L.-H.T. and L.Z. provided resources; L.-Y.X., X.X., N.J., H.-F.W., Y.-S.P., Q.W., S.-F.Z. and L.Z. conducted the investigations; M.S., W.S., R.-Z.Y., J.-J.Z., W.-C.W. and N.J. conducted formal analysis; J.-F.J. and X.-M.C. administered the project; W.S., M.S., J.-J.Z. and R.-Z.Y. curated the data; R.-Z.Y., X.H. and J.-J.Z. created visualizations; W.-C.C., M.S., J.-F.J. and W.S. wrote the original draft; W.-C.C., J.-F.J., W.S. and M.S. reviewed and edited the article; W.-C.C., J.-F.J., M.S. and Y.-L.H. acquired funding; W.-C.C., J.-F.J. and Y.-L.H. supervised the project.

## Competing interests

The authors declare no competing interests.

## Additional information

**Extended data** is available for this paper at https://doi.org/10.1038/s41564-022-01181-1.

**Correspondence and requests for materials** should be addressed to Jia-Fu Jiang, Yan-Ling Hu or Wu-Chun Cao.

**The specific RT-PCR primers designed according to assembled virus sequences in meta-transcriptome data**

| Organism | Primer | Primer sequence (5'-3') | Annealing temperature (℃) | Length of product (bp) |
|---|---|---|---|---|
| Pangolin hunnivirus BIME1 | HU2-F | CAGGCGGAAGTAAGCAAGAAT | 54 | 467 |
| | HU2-R | TCAAGGGAACCCAAGGAAGAG | | |
| Pangolin hunnivirus BIME2 | HUB1-1F | GTCATGGGTTCTGGTGGAAGT | 55 | 255 |
| | HUB1-1R | GCTAAGGCAGCAAAGGGTGTT | | |
| | HUB1-2F | GTGGATTTGGAAGGTGCTTTA | 53 | |
| | HUB1-2R | TGTCGTGCTTACCAGTGTAGAG | | |
| Pangolin hunnivirus BIME3 | HUM1-1F | GGGTGCCGCACGTAGAGGAGAT | 60 | 310 |
| | HUM1-1R | GCAGGCCGCCCACAAGTGAAT | | |
| | HUM1-2F | AAACTAGACCCAATGAGAAGG | 50 | |
| | HUM1-2R | CAACATCATCAAATCCCAACT | | |
| Pangolin hunnivirus BIME4 | HU1-F | CGGGTTACTCCCACTTTCACA | 53 | 481 |
| | HU1-R | TACCAATGCGTTTCATCCACA | | |
| Pangolin hunnivirus BIME5 | HU3-F | GCAGACAGTGTAACGGAAGAT | 53 | 410 |
| | HU3-R | TATAGGAGTCAAGTGCAGGGA | | |
| Pangolin pestivirus BIME1 | PE2-F | TATGCGTCCTATGGCTATTTC | 51 | 407 |
| | PE2-R | TATGTTTCTTATGGCTGTGGG | | |
| Pangolin pestivirus BIME2 | PE4-F | CAGACTCCTCCAGATTCCTAA | 52 | 397 |
| | PE4-R | TTTCCCACCAACTACTCAACT | | |
| Pangolin pestivirus BIME3 | PE3-F | AAACAATGACAGTAATCGGAAGTG | 54 | 394 |
| | PE3-R | ATGTAGGGAATCTGGTGGGAG | | |
| Pangolin pestivirus BIME7 | PE5-F | TCTTGATAAACTGACTGCCTTCT | 53 | 461 |
| | PE5-R | CTGCTTTGACTCTTCCCACTA | | |
| Pangolin pestivirus BIME9 | PE1-F | GAGATAACTAAAGACGGGACA | 51 | 451 |
| | PE1-R | GTAGGAGAAGAAAGTGGGATA | | |
| Pangolin copiparvovirus BIME1 | CO-F | CTCCTGCACAATGGTTATCTG | 55 | 330 |
| | CO-R | ACTAGGTCCACCCTGTCCTCT | | |
| Pangolin respiratory syncytial virus A | HR-F | TGAGACCATTATCGCTTGAGA | 50 | 450 |
| | HR-R | GACTTTGCTAAGAGCCATTTT | | |
| Orthopneumovirus BIME1 | outer primer OP1-1F | AGTCCGACTGGTCCACCTACATC | 50 | 236 |
| | outer primer OP1-1R | GAAGGCAGGAGTGGGTAGCGAAT | | |
| | inner primer OP1-2F | CTTCCACGGTCCACAGTCTCC | 44 | |
| | inner primer OP1-2R | GCGGGTTCGGTCAAGCATTTC | | |
| Pangolin rotavirus A | outer primer rota1-1F | CAAGATTGGCTGATAGATTGC | 45 | 227 |
| | outer primer rota1-1R | CTAGTACGCCTACCTGGGACA | | |
| | inner primer rota1-2F | CTTCAAGACTCGGAATTAGCA | 42 | |
| | inner primer rota1-2R | TAGTACGCCTACCTGGGACAT | | |
| Pangolin orthoreovirus | OR-1F | TGACCCATTAAATTCAGATCCTTTTC | 59 | 564 |
| | OR-R | TCACGCTGACCGTCCTTCCTGTCGC | | |
| | OR-2F | GGAGTCAGAAGCGGATGCATTGGCC | 64 | 352 |
| | OR-R | TCACGCTGACCGTCCTTCCTGTCGC | | |
| Pangolin coronavirus HKU4 | 162pcov_F | ACATAAACATACATTTCTCACCC | 50 | 275 |
| | 162pcov_R | AAAACCACATTACACCACTACAC | | |
| Pangolin phlebovirus BIME1 | PH-F | GACTTCTTGGGTCTGGTTTGT | 52 | 402 |
| | PH-R | TTTGGCTGGTTTAGGAATCTC | | |
| Pangolin protoparvovirus | PR-F | TTACGCTGCTTATCTTCGCTCTG | 56 | 448 |
| | PR-R | TGCTGTGATTTCCACCCATCC | | |
| Pangolin respirovirus | RE-F | CCTTGGAGCAGAGGGCAGATT | 56 | 380 |
| | RE-R | GCCTTTACCGAAGTGCGTGAT | | |
| Pangolin shanbavirus BIME1 | 162shanba_F | CAGCGGTACTGGGTTAACTATCC | 50 | 451 |
| | 162shanba_R | TGTTCTTGTTCTTCCGTTTTTGG | | |

**Extended Data Fig. 1 |** The specific RT-PCR primers designed according to assembled virus sequences in meta-transcriptome data.

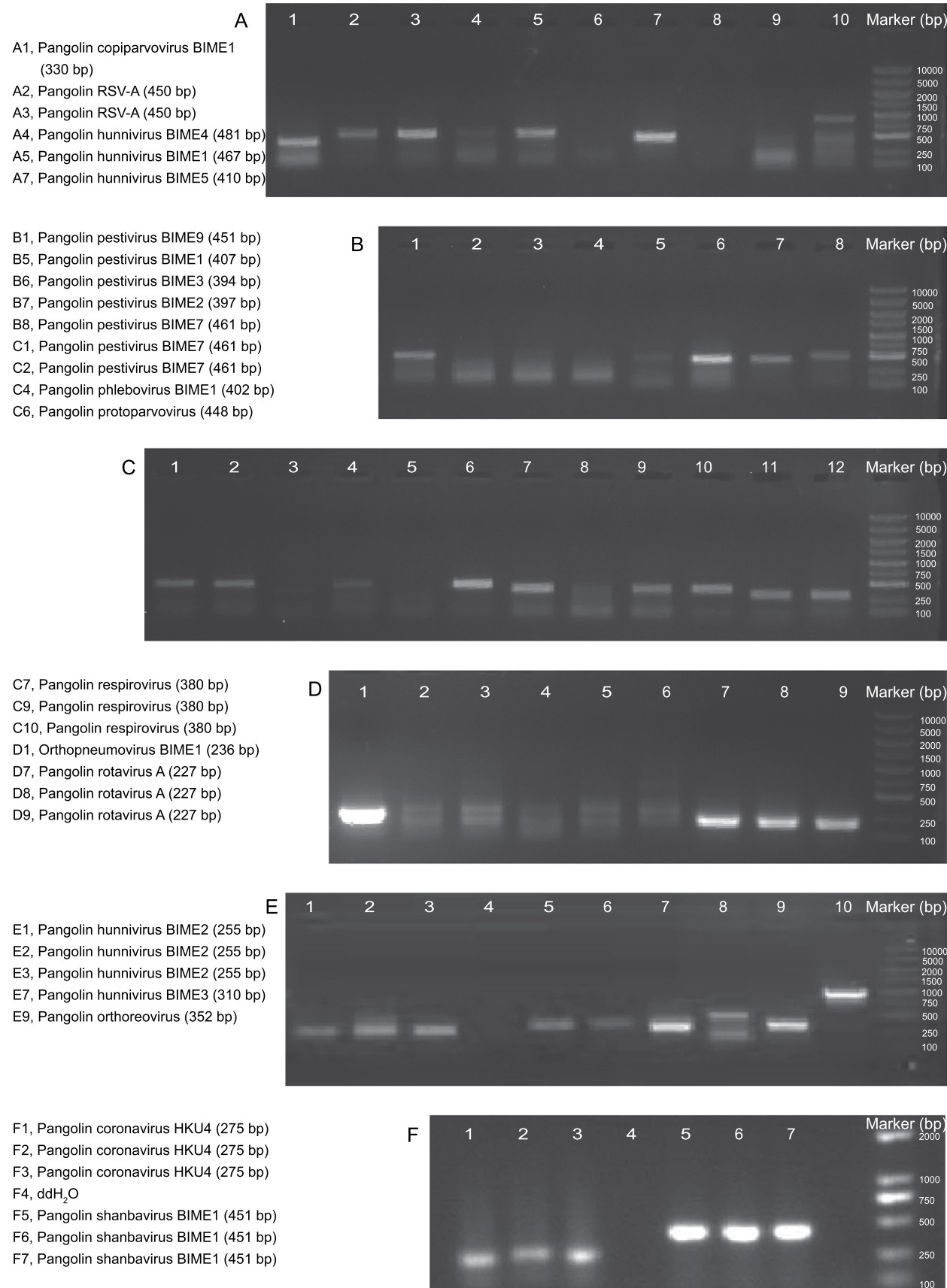

A1, Pangolin copiparvovirus BIME1 (330 bp)
A2, Pangolin RSV-A (450 bp)
A3, Pangolin RSV-A (450 bp)
A4, Pangolin hunnivirus BIME4 (481 bp)
A5, Pangolin hunnivirus BIME1 (467 bp)
A7, Pangolin hunnivirus BIME5 (410 bp)

B1, Pangolin pestivirus BIME9 (451 bp)
B5, Pangolin pestivirus BIME1 (407 bp)
B6, Pangolin pestivirus BIME3 (394 bp)
B7, Pangolin pestivirus BIME2 (397 bp)
B8, Pangolin pestivirus BIME7 (461 bp)
C1, Pangolin pestivirus BIME7 (461 bp)
C2, Pangolin pestivirus BIME7 (461 bp)
C4, Pangolin phlebovirus BIME1 (402 bp)
C6, Pangolin protoparvovirus (448 bp)

C7, Pangolin respirovirus (380 bp)
C9, Pangolin respirovirus (380 bp)
C10, Pangolin respirovirus (380 bp)
D1, Orthopneumovirus BIME1 (236 bp)
D7, Pangolin rotavirus A (227 bp)
D8, Pangolin rotavirus A (227 bp)
D9, Pangolin rotavirus A (227 bp)

E1, Pangolin hunnivirus BIME2 (255 bp)
E2, Pangolin hunnivirus BIME2 (255 bp)
E3, Pangolin hunnivirus BIME2 (255 bp)
E7, Pangolin hunnivirus BIME3 (310 bp)
E9, Pangolin orthoreovirus (352 bp)

F1, Pangolin coronavirus HKU4 (275 bp)
F2, Pangolin coronavirus HKU4 (275 bp)
F3, Pangolin coronavirus HKU4 (275 bp)
F4, ddH$_2$O
F5, Pangolin shanbavirus BIME1 (451 bp)
F6, Pangolin shanbavirus BIME1 (451 bp)
F7, Pangolin shanbavirus BIME1 (451 bp)

**Extended Data Fig. 2 | See next page for caption.**

**Extended Data Fig. 2 | The electrophoretograms of RT-PCR products amplified from pangolin samples.** Each virus identified by MGISEQ-2000 sequencing in this study was tested by a specific RT-PCR assay in the original sample. *Pangolin coronavirus HKU4* and *Pangolin shanbavirus BIME1* were tested by specific RT-PCR three times. The primers used for the RT-PCR tests are listed in Extended Data Fig. 1. To confirm the RT-PCR amplification, all the RT-PCR products were sequenced by Sanger sequencing method. The sequencing diagram for each product is provided in Supplementary Fig. 1.

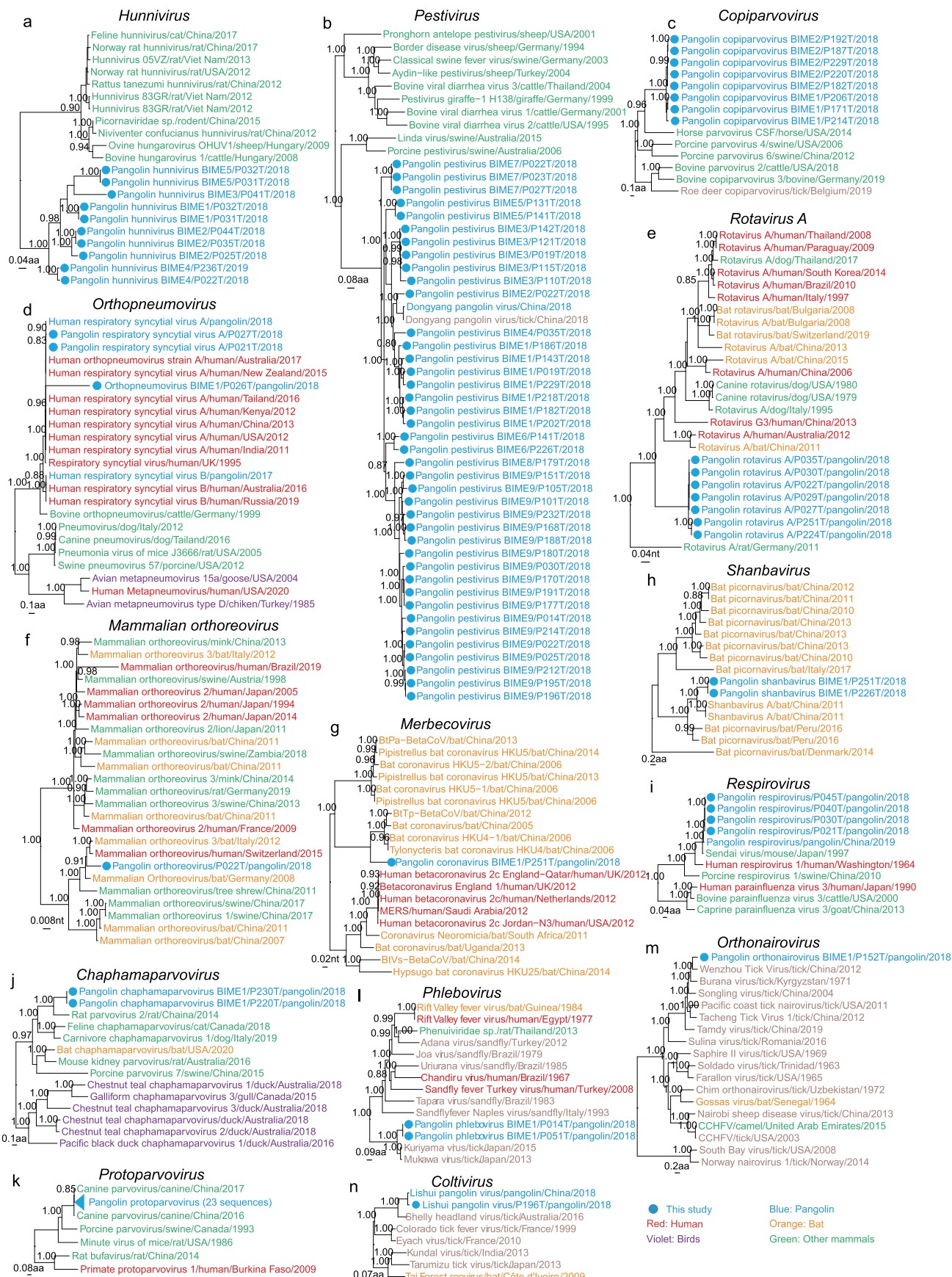

**Extended Data Fig. 3 | Phylogeny of viruses identified in pangolins using the MrBayes.** Phylogenetic analyses by MrBayes use the same sequences and models with the maximum likelihood method.

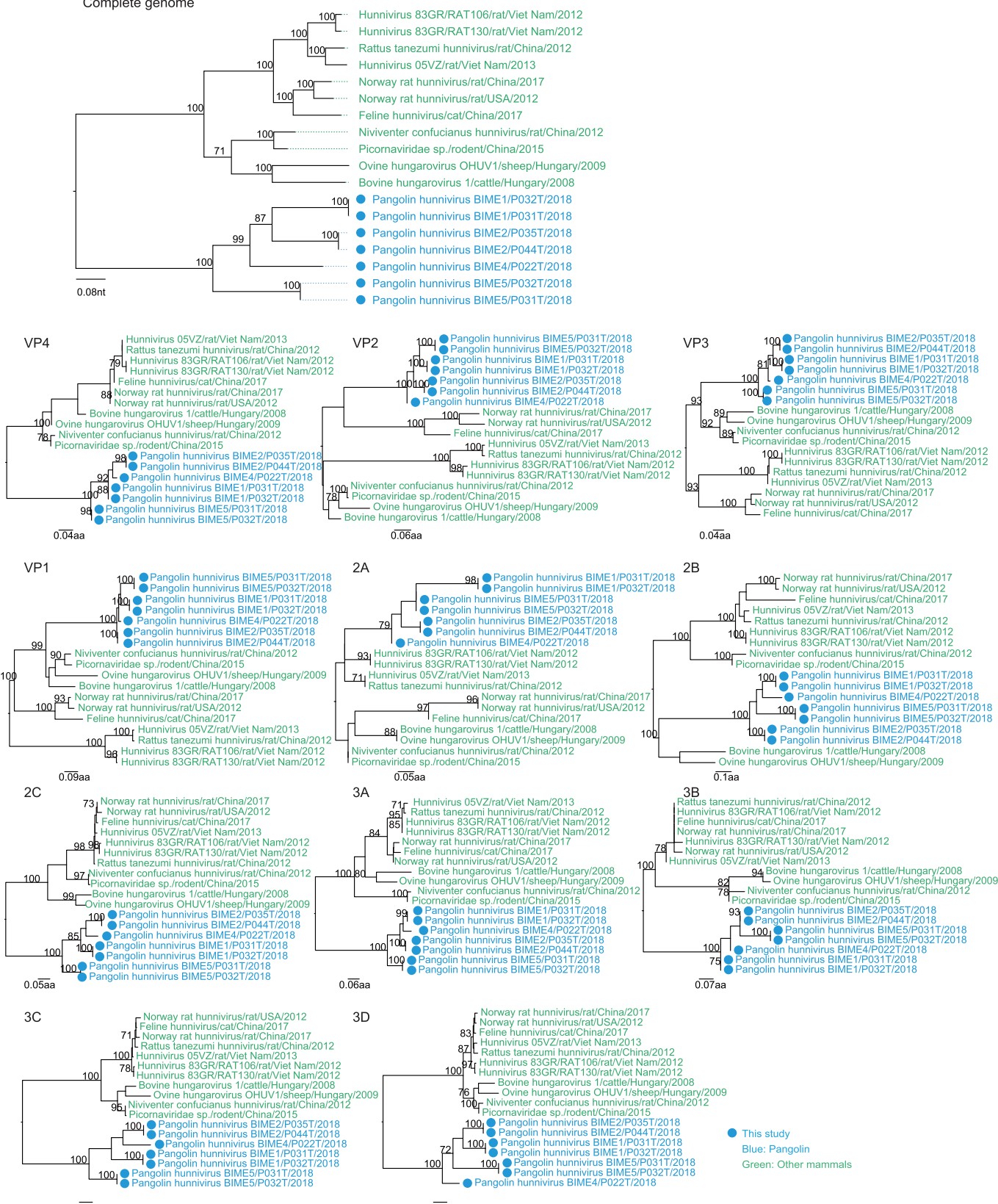

**Extended Data Fig. 4 | Phylogeny of viruses in the *Hunnivirus* genus.** The phylogenetic trees were constructed based on either nucleotide sequences of whole genome or amino acid of 11 regions.

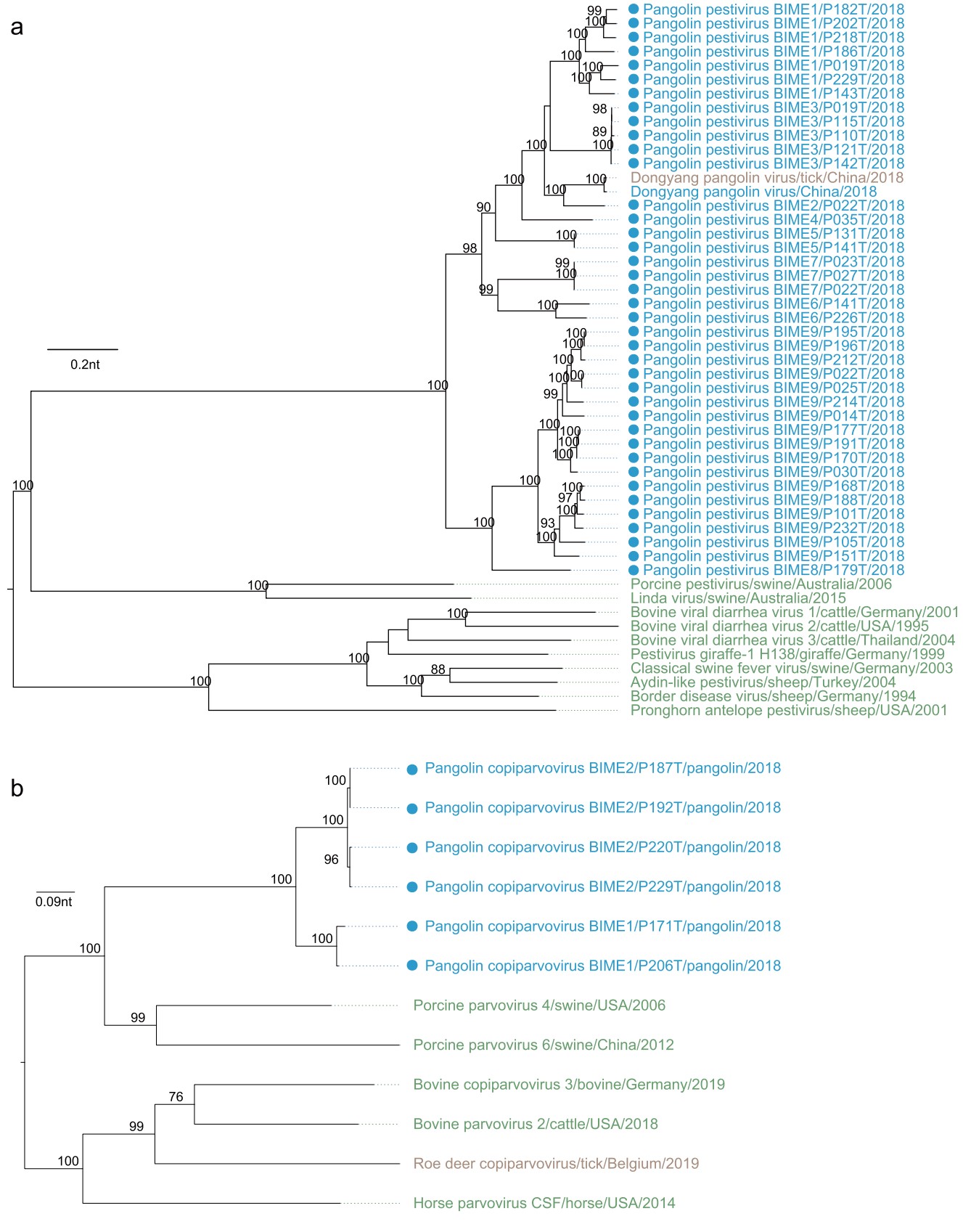

**Extended Data Fig. 5 | Phylogeny of viruses in the *Pestivirus* and *Copiparvovirus* genera based on complete genome sequences. a**, Phylogeny of viruses in the *Pestivirus* genus. **b**, Phylogeny of viruses in the *Copiparvovirus* genus.

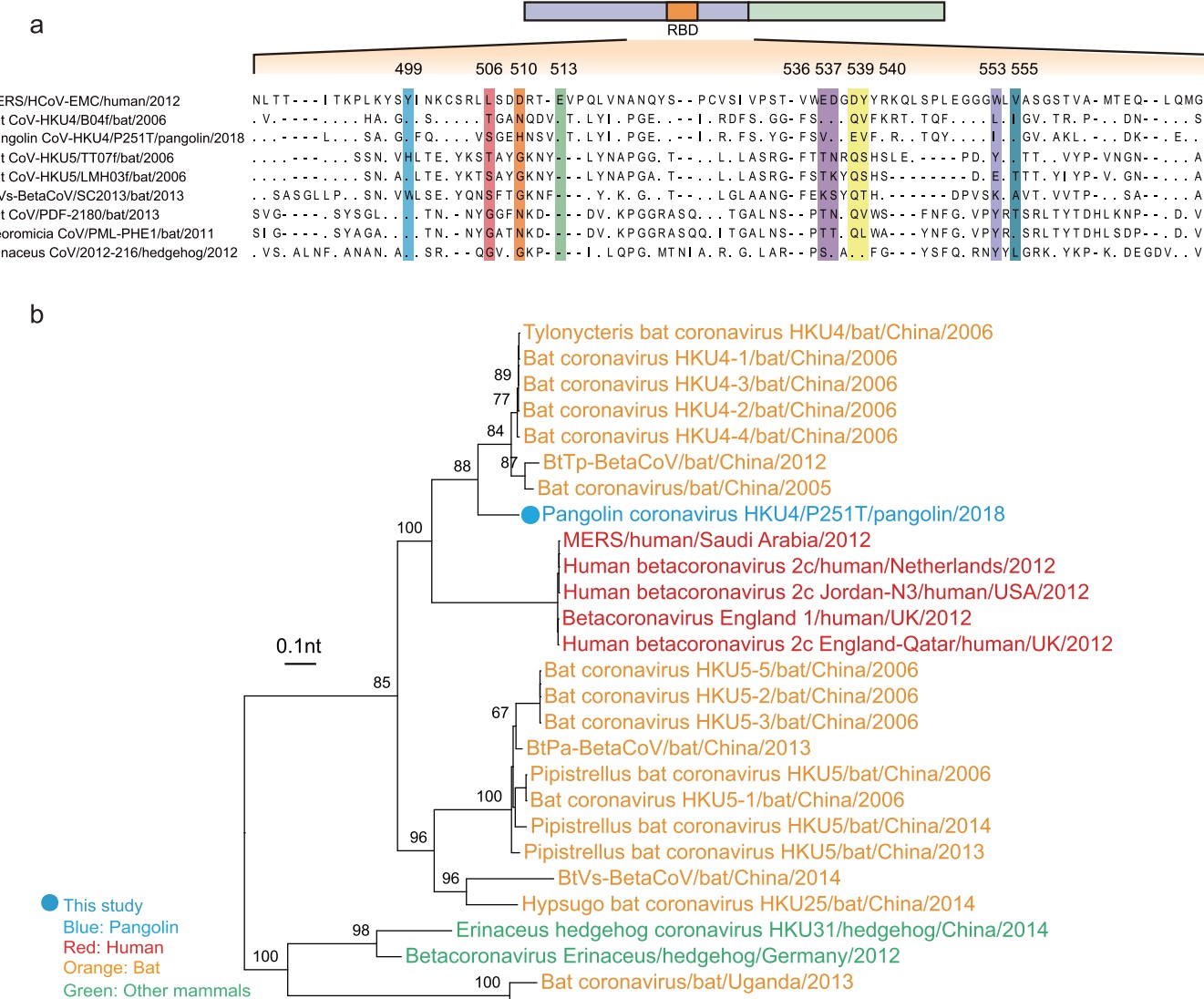

**Extended Data Fig. 6 | Receptor binding domain (RBD) region of *Pangolin coronavirus HKU4*. a**, Sequence alignment of the RBD of viruses in subgenus *Merbecovirus*. The critical residues of RBD are shaded in colors. **b**, Phylogeny of viruses in subgenus *Merbecovirus* based on entire RBD sequences. Tree tips are colored according to host type, using red for humans, blue for pangolins, orange for bats and green for other mammals.

# Reporting Summary

## Statistics

For all statistical analyses, confirm that the following items are present in the figure legend, table legend, main text, or Methods section.

| n/a | Confirmed | |
|---|---|---|
| ☐ | ☒ | The exact sample size (*n*) for each experimental group/condition, given as a discrete number and unit of measurement |
| ☒ | ☐ | A statement on whether measurements were taken from distinct samples or whether the same sample was measured repeatedly |
| ☐ | ☒ | The statistical test(s) used AND whether they are one- or two-sided *Only common tests should be described solely by name; describe more complex techniques in the Methods section.* |
| ☒ | ☐ | A description of all covariates tested |
| ☒ | ☐ | A description of any assumptions or corrections, such as tests of normality and adjustment for multiple comparisons |
| ☒ | ☐ | A full description of the statistical parameters including central tendency (e.g. means) or other basic estimates (e.g. regression coefficient) AND variation (e.g. standard deviation) or associated estimates of uncertainty (e.g. confidence intervals) |
| ☐ | ☒ | For null hypothesis testing, the test statistic (e.g. *F*, *t*, *r*) with confidence intervals, effect sizes, degrees of freedom and *P* value noted *Give P values as exact values whenever suitable.* |
| ☒ | ☐ | For Bayesian analysis, information on the choice of priors and Markov chain Monte Carlo settings |
| ☒ | ☐ | For hierarchical and complex designs, identification of the appropriate level for tests and full reporting of outcomes |
| ☒ | ☐ | Estimates of effect sizes (e.g. Cohen's *d*, Pearson's *r*), indicating how they were calculated |

*Our web collection on statistics for biologists contains articles on many of the points above.*

## Software and code

Policy information about availability of computer code

| Data collection | Reference genome sequence data were downloaded from GenBank using the web interface. |
|---|---|
| Data analysis | Software used: Geneious Prime 2021.2.2, BLAST v2.3.0+, MEGAHIT v1.2.9, MAFFT v7.475, PhyML v3.1, IQTREE v1.6.10, fastp v0.21.0, DIAMOND v0.9.21, Bowtie2 v2.3.4.2, vegan v2.5-7, SimPlot v3.5.1, GATK v4.1.2.0, TrimAl v1.4.rev22, MrBayes v3.2.7 |

For manuscripts utilizing custom algorithms or software that are central to the research but not yet described in published literature, software must be made available to editors and reviewers. We strongly encourage code deposition in a community repository (e.g. GitHub). See the Nature Portfolio guidelines for submitting code & software for further information.

## Data

Policy information about availability of data

All manuscripts must include a data availability statement. This statement should provide the following information, where applicable:

- Accession codes, unique identifiers, or web links for publicly available datasets
- A description of any restrictions on data availability
- For clinical datasets or third party data, please ensure that the statement adheres to our policy

The sequencing reads of 161 pangolins after quality control have been deposited in the NCBI Sequence Read Archive database under the BioProject accession number PRJNA845961. Pangolin mitochondrial sequences and viral sequences assembled in this study were deposited in the GenBank (OM009282-OM009284,OM037454, ON024072-ON024140, ON059801-ON059909, ON166559-ON166563, ON045154-ON045314). We have provided the GenBank accession numbers of each reference sequence in phylogenetic trees files (newick format) and associated sequence alignments (fasta format) available at https://doi.org/10.6084/m9.figshare.19499030.

# Field-specific reporting

Please select the one below that is the best fit for your research. If you are not sure, read the appropriate sections before making your selection.

☒ Life sciences          ☐ Behavioural & social sciences          ☐ Ecological, evolutionary & environmental sciences

For a reference copy of the document with all sections, see nature.com/documents/nr-reporting-summary-flat.pdf

# Life sciences study design

All studies must disclose on these points even when the disclosure is negative.

| | |
|---|---|
| Sample size | No sample size calculation was performed. We collected available archived samples of pangolins smuggled from Southeast Asia to China and confiscated by Customs during 2018-2019. Available tissues (including muscle, lung, intestine, spleen, liver, heart and kidney) of each pangolin were pooled into a single sample. Libraries of 161 pangolin samples were successfully constructed for meta-transcriptome sequencing. |
| Data exclusions | No data were excluded. |
| Replication | We designed specific primers according to assembled virus sequences (Extended Data Fig. 1), and performed RT-PCR to confirm the presence of viruses identified by MGISEQ-2000 sequencing platform (Supplementary Table 1) followed by Sanger sequencing (Supplementary Fig. 1). All the PCR experiments were successful. Viruses identified by NovaSeq 6000 platform were not included in RT-PCR due to lack of sufficient original tissue samples. |
| Randomization | There was no separation of experimental groups in the study, hence no randomization. |
| Blinding | There was no separation of experimental groups in the study, hence no blinding. |

# Reporting for specific materials, systems and methods

We require information from authors about some types of materials, experimental systems and methods used in many studies. Here, indicate whether each material, system or method listed is relevant to your study. If you are not sure if a list item applies to your research, read the appropriate section before selecting a response.

## Materials & experimental systems

| n/a | Involved in the study |
|---|---|
| ☒ ☐ | Antibodies |
| ☒ ☐ | Eukaryotic cell lines |
| ☒ ☐ | Palaeontology and archaeology |
| ☒ ☐ | Animals and other organisms |
| ☒ ☐ | Human research participants |
| ☒ ☐ | Clinical data |
| ☒ ☐ | Dual use research of concern |

## Methods

| n/a | Involved in the study |
|---|---|
| ☒ ☐ | ChIP-seq |
| ☒ ☐ | Flow cytometry |
| ☒ ☐ | MRI-based neuroimaging |

