## [Peer review file · Nature Microbiology]

Peer Review Information

Journal: Nature Microbiology

Manuscript Title: Trafficked Malayan pangolins contain viral pathogens of humans

Corresponding author name(s): Chun Cao

Reviewer Comments & Decisions:

Decision Letter, initial version:

Dear Professor Cao,

Thank you for your patience while your manuscript "Characterization of pangolin virome sheds light on risks of emerging zoonotic and epizootic viruses" was under peer-review at Nature Microbiology. It has now been seen by 3 referees, whose expertise and comments you will find at the end of this email. Although they find your work of some potential interest, they have raised a number of concerns that will need to be addressed before we can consider publication of the work in Nature Microbiology.

In particular, as requested by referee #1, please clearly detail how the sampling was performed and what methods were used, including for the sequencing and bioinformatics analyses. Please also deposit the data in an appropriate repository (see here for a list of accepted repositories: <https://www.nature.com/sdata/policies/repositories>) and make the data available for referees. Please address the concerns from referee #1 and #3 regarding contamination including additional controls. Referee #2 suggests reframing the study and we would encourage you to do this. Referee #2 and #3 have other questions that should be addressed.

Should further experimental data allow you to address these criticisms, we would be happy to look at a revised manuscript.

Please include a data availability statement as a separate section after Methods but before references, under the heading "Data Availability". This section should inform readers about the availability of the data used to support the conclusions of your study. This information includes accession codes to public repositories (data banks for protein, DNA or RNA sequences, microarray, proteomics data etc...), references to source data published alongside the paper, unique identifiers such as URLs to data

2repository entries, or data set DOIs, and any other statement about data availability. At a minimum, you should include the following statement: "The data that support the findings of this study are available from the corresponding author upon request", mentioning any restrictions on availability. If DOIs are provided, we also strongly encourage including these in the Reference list (authors, title, publisher (repository name), identifier, year). For more guidance on how to write this section please see:

<http://www.nature.com/authors/policies/data/data-availability-statements-data-citations.pdf>

* If you have not done so already we suggest that you begin to revise your manuscript so that it conforms to our Article format instructions at <http://www.nature.com/nmicrobiol/info/final-submission>. Refer also to any guidelines provided in this letter.

When submitting the revised version of your manuscript, please pay close attention to our [href="https://www.nature.com/nature-research/editorial-policies/image-integrity">Digital Image Integrity Guidelines. and to the following points below:](https://www.nature.com/nature-research/editorial-policies/image-integrity)

{redacted}

Note: This url links to your confidential homepage and associated information about manuscripts you may have submitted or be reviewing for us. If you wish to forward this e-mail

2to co-authors, please delete this link to your homepage first.

Nature Microbiology is committed to improving transparency in authorship. As part of our efforts in this direction, we are now requesting that all authors identified as 'corresponding author' on published papers create and link their Open Researcher and Contributor Identifier (ORCID) with their account on the Manuscript Tracking System (MTS), prior to acceptance. This applies to primary research papers only. ORCID helps the scientific community achieve unambiguous attribution of all scholarly contributions. You can create and link your ORCID from the home page of the MTS by clicking on 'Modify my Springer Nature account'. For more information please visit www.springernature.com/orcid.

If you wish to submit a suitably revised manuscript we would hope to receive it within 6 months. If you cannot send it within this time, please let us know. We will be happy to consider your revision, even if a similar study has been accepted for publication at Nature Microbiology or published elsewhere (up to a maximum of 6 months).

Yours sincerely,

{redacted}

Reviewer Expertise:

Referee #1: viral genomics, evolution and epidemiology

Referee #2: ecology, epidemiology and evolution of infectious diseases

Referee #3: infectious disease ecology

Reviewer Comments:

Reviewer #1 (Remarks to the Author):

The manuscript by Shi et al. describes a study in which 161 pangolins in China were sampled and sequenced. A number of virus sequences were identified using standard methods. Overall, the methods used to generate the data are poorly described and it is difficult to determine if the resulting virus genome sequences were assembled from single animals or pooled samples. The language is inflated with multiple claims of the identification of novel and pangolin-specific viruses. The manuscript does not provide the careful description of the data and analysis that would be expected from this journal. The authors should consider the following points.

1. The details on the number of tissue samples per animal and total animal samples, and how the samples were pooled are confusing.

Line 78-79, page 5: "Then, we pooled various available tissues (muscle, lung, intestine, spleen, liver,

3heart or kidney) of each pangolin, and successfully constructed 161 libraries for meta-transcriptome sequencing."

This is not clear. Were muscle, lung, intestine spleen, liver heart and kidney from each animal pooled into a single sample, or were all muscle samples, all lung samples etc. pooled. Were the authors successful in collecting all described tissues including muscle, lung, intestine, spleen, liver, heart and kidney for each animal? Or different animals yield different number of tissues collection.

This should be clearly described and also included in the table of genome details (see next point),

2. A table should be included with the following information for each new genome

Source: animal id. if pooled, tissues collected for the animal, length of genome or segment, number of reads mapping to final genome, GenBank accession number. This list of information is the minimal. If there are additional metadata available, the authors should provide the data in this table to allow readers to understand the sample collection and study design better.

3. Line 438-452, page 20, methods on RNA extraction, library preparation and sequencing. Controls for contamination by other virus samples handled in the same sequencing facility should be provided. Data from water controls sequenced in the same sequencing runs should be provided. Details on how the authors controlled for cross-talk carried over from previous sequencing runs should be described. Details of the sequencing runs (how many raw reads obtained per sample, pooling of samples if it was performed) should be provided.

Line 446: "The constructed library was qualified with a Qubit 4.0 fluorometer, as well as with Qsep-100 for library fragment size analysis". A Qubit 4.0 fluorometer can determine the total DNA concentration; it cannot "qualify" the library or to analyse the DNA fragment length. This sentence should be changed to accurately describe the methods that the authors used for what purpose.

4. Line 166-170, page 8. "Phylogenetic analysis revealed that the two RSV-As from pangolins clustered with human RSV-As identified around the world (Fig. 3a). The whole genome sequences of the two RSV-As were identical and had 99.6% similarity with their closest relative RSV-A strain detected in humans from Australia in 201725."

This should raise some alarms about possible contamination. It is surprising that two identical RSV sequences that are so close to a human RSV sequence were found in pangolins. The 100% identity between the two sequences is surprising as well and unfortunately no source details were provided to help understand this identity. Was human RSV handled in the same laboratory? Or was human RSV sequenced in the same sequencer in the previous runs? Some comments on these surprisingly close sequences as well as providing data to rule out possible contamination should be provided. A discussion of the biological plausibility of this result including the known host range of human RSV should be included.

5. Line 35 page 3, Abstract "and discovered 29 distinctive vertebrate-associated viruses" this overstates the results. Sequences from these viruses were identified, the virus was not "discovered". This should be reworded.

6. Line 43-44 page 3, Abstract "Overall, these vertebrate-associated viruses distributed in pangolins regardless of pangolin genetic groups."

This statement "regardless of pangolin genetic groups" is contradicted in the text by the statement

4that all of the pangolins were from one species *Manis javanica* – line 80-82 Page 5: "Considering lack of species classification of pangolins, we identified their species based on mitochondrial contigs in the meta-transcriptome data, and found all 161 individuals belonged to *Manis javanica*". Only one of these statements can be correct.

7. Line 184-186 page 9: "rotavirus A might evolve in pangolins for a relatively long time, and pangolins might be potential reservoirs for new strains rather than simply spill-over hosts." I believe the authors mean "might have been evolving". However, the simple presence of a viral sequence does not mean that the virus has been replicating in this host or evolving for a "relatively long time". I don't see any time data or actual evidence of virus replication. This kind of statements without supporting data or evidence weaken the manuscript and any conclusions the authors would like to make.

Line 179-180: "The sequences of 11 segments had 93.37%-100% nt identities with each other". As currently written, it would mean that 11 segments of the rotavirus A shared 93.37%-100% nt identities with each other, which is impossible!

8. Throughout the document, the authors use "virus" when more correctly they are discussing "virus sequences". No actual viruses were isolated. It is misleading to say that a virus was identified or that a virus was present or a virus evolved in this species. The authors should not misrepresent their findings in this way and the text corrected.

Examples (a few of many):

Line 102, page 6 "Despite being present in pangolin libraries, viruses that are likely ..."

Line 214-215, page 10: "The presence of tick-associated viruses.."

Line 220-221, page 10: "We identified a HKU4 coronavirus strain P251T..."

Line 238, page 11: "A virus in genus *Respirovirus* of family *Paramyxoviridae* was detected..."

Line 249-250, page 11: "These diverse mammal-associated viruses imply multiple cross-species transmission events between pangolins and other mammals."

Line 252, page 12 "Virus distribution in genetic groups of pangolins."

Line 470, page 21 " If a newly discovered virus..."

Figure 3 "Fig. 3: Known human pathogens in pangolins."

These were sequences, not actual viruses detected and the text should be modified to correctly communicate this without misleading the reader.

9. Line 80-82, page 5: "Considering lack of species classification of pangolins". It is not clear what this means. The species classifications of the pangolins tested were not known or there is no species classification for pangolin? This should be clarified.

10. Line 110, page 6: "A total of 17 novel viruses". Please define "novel". If the sequences were identified as *Hunnivirus*, *Pestivirus* and *Copiparvovirus* sequences, then these are not really novel. The wording should be revised or a definition of "novel" should be provided. I would avoid using the term novel unless the authors have truly found sequences that do not classify in any known virus family. Also, again the confusion between identifying a virus and reporting a virus sequence should be

5addressed.

11. Line 119-121, page 6: "Phylogenetic analysis based on RdRp showed that all the five novel species formed a new cluster distinct from all previously known species in genus Hunnivirus (Fig. 2a)."

Same concern as above point, do these sequences provide evidence of a novel species? The criteria for Picornavirus species are quite well defined and the authors need to describe these sequences correctly without hype.

12. Line 521-525, page 23, Data Availability. "Sequences assembled in this study were deposited in the National Omics Data Encyclopedia (NODE) (<http://www.biosino.org/node>) (OEZ008244) and are publicly available as of the date of publication. Accession numbers are listed in Extended Data Table 3."

In Extended Data Table 3. the authors say the data are in a repository. However, no accession numbers are provided. The assembled genomes should be deposited in GenBank and accession numbers should be provided instead of saying "TBD".

13. Line 305-307, page 14: "The first limitation of the study is pooling different tissues of pangolins for metagenomic sequencing, from which the distribution of viruses in different organs cannot be observed."

The danger of pooling is that virus sequence chimeras can be generated if the individual animals were infected with different but closely related viruses. This should be considered and noted in the analysis that the apparent diversity may be an artefact of pooling.

14. Line 299-300, page 13: "In addition, we found pangolin orthoreovirus contain recombination sequence fragments from bat strains." It should be commented that the odd sequence could be an artefact of assembly from multiple related viruses. Without information of the pooling, it is difficult to assess this validity of these sequences.

15. The use of novel throughout the manuscript is misleading.

For example, Extended data table 1 "novel/known: column,

All of these viruses are known, what are the criteria for calling a hunivirus or pestivirus novel?

Line 36-38, page 3: "Seventeen novel viruses in Hunnivirus, Pestivirus and Copiparvovirus genera formed well-supported monophyletic clusters, which might be pangolin-specific."

Again, no criteria for calling a sequence (not a virus) novel.

Also line 285-286, page 13: "First, we identified 17 pangolin-specific novel viruses in Hunnivirus, Pestivirus and Copiparvovirus genera,"

The statement that these might be pangolin specific is also misleading. Such a result is highly dependent on the rest of the world sampling and simply finding a monophyletic cluster in a sparsely sampled virus family does not make the virus host specific. This statement should be revised carefully.

16. Supplementary Table 1. It is not clear what the values represent, perhaps amino acid length? This should be specified in a table legend.

"Range of similarity variation compared to NC_023176.1" ♦ The method of calculating this value

6should be indicated.

The GenBank accession number for each sequence (including the new sequences reported in the manuscript) should be provided.

17. Figure 1. The trees are difficult to read, the authors should decide on the phylogenetic feature they are hoping to illustrate with each tree and prepare an appropriate subset of the reference sequences. For example, the picornavirus tree is just a swarm of lines with a blue box in the middle. Was it necessary to include so many Caliciviridae and Picornavirales sequences in one tree?

18. The use of the term epizootic is misleading. The word in standard usage means broadly infecting a species and it is difficult to conclude from sampling of a small set of pangolins in one area that any of the virus sequences indicate an epizootic infection. This term should be removed as it is misleading.

Reviewer #2 (Remarks to the Author):

This manuscript describes a study of the virome of Malayan pangolins confiscated from the illegal wildlife trade in China. The authors identify a large number of viruses, some that form new clades within known viral families and thus appear to be pangolin-specific, and others that cluster with non-pangolin viruses and appear to be recently transmitted to pangolins from other species. The topic is timely, given suspicions that pangolins were somehow involved in the early transmission of SARS-CoV-2 (I believe these hypotheses have largely fallen by the wayside).

The analyses are very well done. In particular, the “forest” of phylogenetic trees presented speaks to the skills of this research group in phylogenetic analyses. The figures are presented clearly, and the figures and other data support the phylogenetic inferences within the paper.

My main critique is not with the data or analysis, but rather with the organization and interpretations. These pangolins were confiscated from the illegal wildlife trade, and the conditions of their capture and transport are unknown. The authors mention that some of the viruses they found could have been transmitted to pangolins from other species, but this idea is not discussed very much. This point should, in my opinion, be central to the manuscript.

As it stands, the manuscript is a “litany of viruses” paper (i.e. “Look at all the viruses we found!”). However, the study would be more interesting and more important if it were framed as follows: Smuggled Malaysian pangolins harbor some viruses that look like “pangolin viruses” and other viruses that look like they were transmitted from other species, including humans. In other words, there are two types of viruses here: pangolin viruses, and non-pangolin viruses. This distinction is buried in the text. It should be a main focus, from the abstract to the discussion. This is important because it is widely believed is that mixing together of animals like pangolins with other animals like bats and civets (both identified as viral hosts in the paper) is a major risk factor for disease emergence. This is

7the “wet market” hypothesis – that the conditions of wildlife transport and sale create conditions that favor cross-species transmission of viruses. These authors have found compelling evidence in support of that hypothesis, and this evidence is, in my opinion, their most important finding.

Finally, all the pangolins were one species. The authors imply in several places that they have studied the “pangolin virome,” but it is really the “Malayan pangolin virome.”

In light of these points, the authors might consider a title such as: “Virome of illegally trafficked Malayan pangolins reveals novel pangolin-specific viral lineages and infection with known pathogens of humans.”

Specific comments

Page 4, line 24: Should “gamey meat” be changed to “wild game?” I think this may be an issue with translation.

Page 5, lines 4-6. It is unclear what is meant by “lack of species classification of pangolins.” Are the authors referring to difficulty with morphological identification of pangolin species, or to lack of DNA sequences from pangolins of known species in public databases? Also, the mitochondrial contigs should be deposited into a public database. I suspect there was variability among pangolins in contig length and mitogenome coverage, so species assignment criteria (e.g. E-values) should be provided.

Page 7, lines 1-5. It is very interesting that the L-protein has been lost from these viruses is an important and interesting finding. If space permits, consider mentioning this in the Abstract, because it increases confidence in the ability to detect pangolin-adapted viruses.

Page 7, lines 11-13. Many pestiviruses appear to be benign. The authors should not over-emphasize the pathogenicity of pestiviruses as a genus.

Page 8, line 20. The novel Orthopneumovirus is interesting, but it could be an unknown human variant or a virus or from another species that pangolins have encountered during smuggling. It would be better to call this virus something other than “pangolin orthopneumovirus” in case it is identified as actually being a virus of another species.

Page 9, lines 11-17. The pangolin-civet connection screams of wildlife trafficking and “wet markets.” Pangolins and civets, along with bats, are the most often suspected reservoirs or amplifying hosts of viruses when they are mixed together. The authors should be sure to emphasize this fact (see suggestion above for reframing the paper).

Page 10, lines 14-15. Ticks can be highly host specific, so the risk of cross-species transmission might not be very high.

Page 12, lines 6-7. It is unclear what is meant here by “sufficient number of mitochondrial variants.”

Page 12, line 25. “Protected” should be used in place of “preserved.”

Page 13, line 8. The conclusion that “Our findings indicate that pangolins harbor diverse viruses with zoonotic and epizootic risks.” Is an overstatement and not very novel. With the alternative framing suggested above (pangolins harbor a mix of pangolin viruses and non-pangolin viruses), the authors could write a more compelling final paragraph. An important point is that the non-pangolin viruses were likely acquired during smuggling, in support of the “pangolin-bat-civet” hypothesis of viral emergence in captured wildlife and in markets.

Lines 14-18. From what I know, there has been no documented case of a pangolin virus being transmitted to humans or to cause human epidemics. It would be good to tone this statement down. However, the authors should definitely keep the conclusion that pangolins could infect humans. Hopefully, that will deter poachers (but I doubt it, unfortunately).

Page 21, lines 11-14. What PCR assays were used to verify the three viruses?

Page 22, line 11. “Determine” is too strong here. Choose a word that reflects the uncertainty inherent in phylogenetic analyses.

Page 23, line 11. It would be good to deposit the assembled viral and pangolin mtDNA sequences in GenBank so that they will appear in BLAST searches at NCBI. The authors relied heavily on NCBI for their analyses, so they should deposit their sequences in NCBI too.

Figure 3. The rotavirus A tree implies that the infected pangolins might have been infected from a common source. Perhaps these 7 pangolins were kept in the same cage and contracted rotavirus from the same person, perhaps also transmitting it to each other. The authors should discuss such possibilities, because these sorts of observations are what one would expect from the horrific conditions in which trafficked wildlife are kept.

Reviewer #3 (Remarks to the Author):

Shi and colleagues present work, titled “Characterization of pangolin virome sheds light on risks of emerging zoonotic and epizootic viruses” that uses meta-transcriptome sequencing to begin to characterize the viromes of pangolin. Pangolins were, until recently, not present in the virology literature, but the discovery of SARS-CoV-2-related coronaviruses in Malayan pangolins in China has highlighted the need for further work. Here authors investigate the virome using samples from 161 pangolins smuggled into China. They discovered number virus genomes or genome fragments and report 29 distinctive vertebrate-associated viruses, with 22 novel viruses. Their work included four “human-associated viruses”, including respiratory syncytial virus, a novel orthopneumovirus, rotavirus A and mammalian orthoreovirus, tick-associated and mammal-associated viruses, including coronavirus closely related to HKU4-CoV from bats. There suggest the discovery of diverse viruses means that pangolins should be considered potential reservoirs or intermediate hosts of zoonotic or epizootic pathogens.

Overall, I think the work is valuable, well performed and presented, the arguments clear and the literature well used. Pangolin are, as the authors report, largely unexplored hosts of viruses (or other

9infectious diseases). The work here is the natural next step, given previous work on coronaviruses and will be a useful addition to the field. The species sampled and diversity of viruses make it an interesting study. I don't use all the tools (especially the bioinformatics), but the methods used seem standard and the data and methods are provided and therefore repeatable. The results provide many hypotheses for future work and the authors are, I believe, suitably cautious with their interpretations. Here are some comments (not in order of importance).

Throughout the manuscript, unless the journal has its own formatting, the International Committee on Viral Taxonomy formatting for viral nomenclature should be followed. Throughout it seems correct for mammals (for example, class and order level taxa names not italicised) but ICVT recommended italicisation for all (e.g. see online at ICVT FAQs:

<https://talk.ictvonline.org/information/w/faq/386/how-to-write-virus-species-and-other-taxa-names>).

Title/line 1: Insert: "Malayan" to the pangolin name, as there is only one species (*Manis javanica*) reported.

Abstract/line 42: "...HKU4-CoV discovered in bats...?"

Intro/line 56: I believe the demand for pangolin is largely from Asia, but the authors are right that N America is involved with the trade, based on the literature, but the text is slightly misleading as written. I would rephrase this and use peer-reviewed literature to clarify these relationships, e.g. Heinrich et al, *Global Ecology & Conservation*, 2016 "Where did all the pangolins go? International CITES trade in pangolin species".

Line 90/Fig 1a: I agree with these statement regarding 'vertebrate-associated' viruses, but I would like some clarification around some of the viruses and their host associations. In Fig 1a (and elsewhere) some of the vertebrate host associations are likely, and evidence strong, even for viral families associated with a range of hosts, but for some it's not clear. For example, in Fig 1a, the rhabdovirus sequences are classified as vertebrate, but the Rhabdoviridae family has a wide host association and not all are vertebrate associated and it's not clear from the tree in Fig 1b which rhabdoviruses the new sequences are most close to. Can the authors review and revise/confirm the putative viral-host relationships?

Lines 162/311: A general point, with respect to viruses associated with humans, how sure can the authors be that there was not simply contamination of samples with human viruses/tissues, rather than true infection? For example, the Human respiratory syncytial virus subtype A (RSV-A) data is convincing (including because of the pangolin orthopneumovirus BIME 1) but where the authors write: "...we cannot trace their original infection sites of the viruses", this can include a comment on contamination of samples, or more details in the methods about how this might have been mitigated? Given the source (trafficked animals) and how these might have been collected it seems contamination rather than infection (viral replication) must at least be acknowledged.

Lines 184-186: I agree (I think), but perhaps revise this sentence to: "These findings suggest that rotavirus A might have transmitted long enough among pangolin for viral evolution to lead to new genotypes, and pangolins might"

Line 193: This might be worth discussing very briefly, because recombinant corona-orthoreoviruses have been observed in bats (<https://www.mdpi.com/1999-4915/12/5/539/htm>)

Line 195: "Pangolin were easily..." implies they were infected here. Maybe revise this to "...were previously reported to be heavily infested with ticks ..." or "...may be heavily infested with ..."

Line 281: I agree, this is a good number of samples, but honestly, pangolin confiscations have included up to 120 tonnes of whole of pangolin, there are ample opportunities for more sample collection. Though hopefully this will cease, perhaps the authors could make a recommendation that

10sample collection is included in animal seizures?

Line 470/471: This seems fair, but not that ICVT has classification processes, which the authors used for the rotavirus discussion. Did the authors look at all viruses using ICVT thresholds?

Line 501-502: did the authors look at ambiguously aligned regions? (And if so, consider trimming, e.g. TrimAI, and realignment)?

Line 504: my guess is it won't change the overall findings, given the tree topologies and branch supports, but did the authors try another approach, e.g. MrBayes to confirm the topologies, especially for key relationships?

Fig 6: can you increase the circle and point size?

Author Rebuttal to Initial comments

Responses to reviewer 1:

The manuscript by Shi et al. describes a study in which 161 pangolins in China were sampled and sequenced. A number of virus sequences were identified using standard methods. Overall, the methods used to generate the data are poorly described and it is difficult to determine if the resulting virus genome sequences were assembled from single animals or pooled samples. The language is inflated with multiple claims of the identification of novel and pangolin-specific viruses. The manuscript does not provide the careful description of the data and analysis that would be expected from this journal. The authors should consider the following points.

[Response]: We appreciate the reviewer's comments, and have described methods used to generate the data in detail as response to comment #1. We also revised the statements regarding claims of the identification of novel and pangolin-specific viruses, and provide related responses below. As suggested by the reviewer, we have provided detailed description of the data analysis in the revised manuscript.

1. The details on the number of tissue samples per animal and total animal samples, and how the samples were pooled are confusing.

Line 78-79, page 5: "Then, we pooled various available tissues (muscle, lung, intestine, spleen, liver, heart or kidney) of each pangolin, and successfully constructed 161 libraries for meta-transcriptome sequencing."

This is not clear. Were muscle, lung, intestine, spleen, liver, heart and kidney from each animal pooled into a single sample, or were all muscle samples, all lung samples etc. pooled. Were the authors successful in collecting all described tissues including muscle, lung, intestine, spleen, liver, heart and kidney for each

11animal? Or different animals yield different number of tissues collection. This should be clearly described and also included in the table of genome details (see next point),

[Response]: In response to this inquiry, we have clarified the pooling procedure in the text as following:

“We collected various types of archived tissues (including muscle, lung, intestine, spleen, liver, heart and kidney), and pooled the tissues from each pangolin as a single sample. The libraries of 161 pangolin samples were successfully constructed for meta-transcriptome sequencing. Detailed information of each library was listed in Supplementary Table 1.” (Page 5, Line 81-85)

2. A table should be included with the following information for each new genome

Source: animal id. if pooled, tissues collected for the animal, length of genome or segment, number of reads mapping to final genome, GenBank accession number. This list of information is the minimal. If there are additional metadata available, the authors should provide the data in this table to allow readers to understand the sample collection and study design better.

[Response]: Following the suggestion, we have added Supplementary Table 1 and Supplementary Table 2. Supplementary Table 1 provides the sample ID, tissues pooled in each sample, number of reads after quality control, and mitochondrial sequence of each pangolin with GenBank accession number. Supplementary Table 2 contains essential information of each viral sequence, including sample ID, name of virus, length of genome or segment, number of reads mapping to final genome, and GenBank accession number.

3. Line 438-452, page 20, methods on RNA extraction, library preparation and sequencing. Controls for contamination by other virus samples handled in the same sequencing facility should be provided. Data from water controls sequenced in the same sequencing runs should be provided. Details on how the authors controlled for cross-talk carried over from previous sequencing runs should be described. Details of the sequencing runs (how many raw reads obtained per sample, pooling of samples if it was performed) should be provided.

[Response]: We appreciated the valuable comments, and have provided detailed description for controlling possible contamination during library preparation and sequencing in the revised manuscript. In the Materials and Methods section, we added “To identify and eliminate possible sequencing or reagent contaminants, we included a sterile water as well as reagent mix as controls⁵¹. The controls were sequenced in the same chip with the libraries of pangolin samples.” (Page 22, Line 498-501)

In addition, methods to control cross-talk contaminant was described as following: “We compared the presence of viral sequences and their abundance across sequencing libraries. In case the identical or near-identical viral sequences were observed in multiple libraries, we used the following steps to rule out possible contamination. We first estimated the read ratio between the highest abundance library and other lower abundance libraries in the same chip. If the ratio was below the index-hopping rate of the sequencing platforms, the reads in low abundance library were considered as cross-contamination during library preparation and excluded from further analysis⁵¹. To eliminate possible index-hopping, we chose the highest percent of 0.1% as the threshold^{59,60} in the study. To further verify the presence of identified virus sequences rather than from contamination, we performed RT-PCR assays in the available samples of the detected viruses in meta-transcriptome sequencing. The specific RT-PCR primers designed according to assembled virus sequences are listed in Extended Data Table 1. The target RT-PCR products were validated by Sanger sequencing (Supplementary Fig. 1). For *Mammalian orthoreovirus*, RT-PCR was also performed to fill gaps in L1 segment which contained RdRp.” (Page 24, Line 544-559).

As suggested by the reviewer, we have provided detailed information concerning sample ID, pooled tissues for library construction, and reads obtained per sample in Supplementary Table 1 in the revised manuscript.

Line 446: “The constructed library was qualified with a Qubit 4.0 fluorometer, as well as with Qsep-100 for library fragment size analysis”. A Qubit 4.0 fluorometer can determine the total DNA concentration; it cannot “qualify” the library or to analyse the DNA fragment length. This sentence should be changed to accurately describe the methods that the authors used for what purpose.

[Response]: As suggested by the reviewer, we have revised the sentence as: “The RNA quantity of each constructed library was measured using Qubit 4.0 fluorometer. The library fragment length was estimated using Qsep-100 (Hangzhou Houze Biotechnology Co., Ltd.)” (Page 22, Line 493-495).

4. Line 166-170, page 8. "Phylogenetic analysis revealed that the two RSV-As from pangolins clustered with human RSV-As identified around the world (Fig. 3a). The whole genome sequences of the two RSV-As were identical and had 99.6% similarity with their closest relative RSV-A strain detected in humans from Australia in 201725."

This should raise some alarms about possible contamination. It is surprising that two identical RSV sequences that are so close to a human RSV sequence were found in pangolins. The 100% identity between the two sequences is surprising as well and unfortunately no source details were provided to help understand this identity. Was human RSV handled in the same laboratory? Or was human RSV sequenced in the same sequencer in the previous runs? Some comments on these surprisingly close sequences as well as providing data to rule out possible contamination should be provided. A discussion of the biological plausibility of this result including the known host range of human RSV should be included.

[Response]: We understand the reviewer's concern. As described in the Materials and Methods in the revised manuscript, we employed multiple approaches to rule out possible contamination. Furthermore, we confirmed the presence of RSV-A in the two samples using RT-PCR followed by Sanger sequencing. We have added the RT-PCR results and biological plausibility in the revised manuscript as following: "We confirmed the presence of RSV-A in the two samples using RT-PCR followed by Sanger sequencing (Extended Data Fig. 4 and Supplementary Fig. 1). RSV was first identified in chimpanzees and later documented to be a mainly human pathogen²⁸. The identical RSV-A sequences suggest that these two pangolins might be infected from the same infected person or transmitted virus from each other". (Page 9, Line 179-183)

5. Line 35 page 3, Abstract "and discovered 29 distinctive vertebrate-associated viruses" this overstates the results. Sequences from these viruses were identified, the virus was not "discovered". This should be reworded.

[Response]: Many thanks for the suggestion. We have reworded the statement as "and obtained sequences of 28 vertebrate-associated viruses" (Page 3, Line 36-37)

6. Line 43-44 page 3, Abstract "Overall, these vertebrate-associated viruses distributed in pangolins regardless of pangolin genetic groups."

This statement "regardless of pangolin genetic groups" is contradicted in the text by the statement that all of the pangolins were from one species *Manis javanica* – line 80-82 Page 5: "Considering lack of species classification of pangolins, we identified their species based on mitochondrial contigs in the meta-transcriptome data, and found all 161 individuals belonged to *Manis javanica*". Only one of these statements can be correct.

[Response]: We are grateful for the comment. The original statement means: "Overall, these vertebrate-associated viruses distributed in Malayan pangolins regardless of pangolin populations inferred from mitochondrial variants.". Due to space limitation, this statement has been removed from the abstract in the revised manuscript.

7. Line 184-186 page 9: "rotavirus A might evolve in pangolins for a relatively long time, and pangolins might be potential reservoirs for new strains rather than simply spill-over hosts."

I believe the authors mean "might have been evolving". However, the simple presence of a viral sequence does not mean that the virus has been replicating in this host or evolving for a "relatively long time". I don't see any time data or actual evidence of virus replication. This kind of statements without supporting data or evidence weaken the manuscript and any conclusions the authors would like to make.

[Response]: We appreciate the reviewer's suggestion, and have revised this sentence to: "These findings suggest that *Rotavirus A* might have associated with pangolin for an extensive period of time, alternatively, it could be due to lack of sampling that covering the diversity gap between pangolin rotaviruses and those identified from other mammalian host" (Page 9, Line 201-204)

Line 179-180: "The sequences of 11 segments had 93.37%-100% nt identities with each other". As currently written, it would mean that 11 segments of the rotavirus A shared 93.37%-100% nt identities with each other, which is impossible!

[Response]: In response to the comment, we have clarified the statement as: “Of the 11 genomic segments, assembled viral sequences of the same segment showed 93.37%-100% nt identities, and 52.0%-87.7% nt identities with the known strains.” (Page 9, Line 196-198)

8. Throughout the document, the authors use “virus” when more correctly they are discussing “virus sequences”. No actual viruses were isolated. It is misleading to say that a virus was identified or that a virus was present or a virus evolved in this species. The authors should not misrepresent their findings in this way and the text corrected.

Examples (a few of many):

Line 102, page 6 "Despite being present in pangolin libraries, viruses that are likely ..."

Line 214-215, page 10: "The presence of tick-associated viruses.."

Line 220-221, page 10: "We identified a HKU4 coronavirus strain P251T..."

Line 238, page 11: "A virus in genus Respirovirus of family Paramyxoviridae was detected..."

Line 249-250, page 11: "These diverse mammal-associated viruses imply multiple cross-species transmission events between pangolins and other mammals."

Line 252, page 12 "Virus distribution in genetic groups of pangolins."

Line 470, page 21 " If a newly discovered virus..."

Figure 3 "Fig. 3: Known human pathogens in pangolins."

These were sequences, not actual viruses detected and the text should be modified to correctly communicate this without misleading the reader.

[Response]: Following the reviewer’s suggestion, we have replaced ‘virus’ with ‘virus sequence’, ‘sequence of virus’ as well as ‘viral diversity’ whenever appropriate throughout the revised manuscript, especially those mentioned by the reviewer:

Line 102, page 6 "Despite being present in pangolin libraries, virus that are likely ..." has been modified to “Despite being present in pangolin libraries, virus sequences that are likely ..” (Page 6, Line 108)

Line 214-215, page 10: "The presence of tick-associated viruses.." has been modified to “The presence of tick-associated virus sequences.” (Page 13, Line 283)

Line 220-221, page 10: "We identified a HKU4 coronavirus strain P251T..." has been modified to "We obtained the sequence of a HKU4 coronavirus strain P251T..." (Page 10, Line 226-227)

Line 238, page 11: "A virus in genus *Respirovirus* of family *Paramyxoviridae* was detected..." has been modified to "Four sequences of a virus in genus *Respirovirus* of family *Paramyxoviridae* were detected..." (Page 11, Line 246-247)

Line 249-250, page 11: "These diverse mammal-associated viruses imply multiple cross-species transmission events between pangolins and other mammals." has been modified to "These sequences of diverse mammal-associated viruses imply multiple cross-species transmission events between pangolins and other mammals." (Page 12, Line 259-260)

Line 252, page 12 "Virus distribution in genetic groups of pangolins." has been modified to "The viral diversity in relation to Malayan pangolin populations." (Page 13, Line 286)

Line 470, page 21 "If a newly discovered virus..." has been modified to "If a detected virus sequence..." (Page 23, Line 522)

Figure 3 "Fig. 3: Known human pathogens with assembled sequences in pangolins." has been modified to "Fig. 3: Analyses of human-associated virus sequences." (Page 34, Line 699)

9. Line 80-82, page 5: "Considering lack of species classification of pangolins". It is not clear what this means. The species classifications of the pangolins tested were not known or there is no species classification for pangolin? This should be clarified.

[Response]: As suggested, we have clarified the description as: "Because only archived tissue samples were available, it was impossible for us to identify the pangolin species through traditional morphological classification. We then identified pangolin species based on mitochondrial contigs in the meta-transcriptome data, and found all 161 pangolins were *Manis javanica*." (Page 5, Line 85-88)

10. Line 110, page 6: "A total of 17 novel viruses". Please define "novel". If the sequences were identified as Hunnivirus, Pestivirus and Copiparvovirus sequences, then these are not really novel. The wording should be revised or a definition of "novel" should be provided. I would avoid using the term novel unless the authors have truly found sequences that do not classify in any known virus family. Also, again the confusion between identifying a virus and reporting a virus sequence should be addressed.

[Response]: Thanks for the comment. In fact, we define "novel" at species level rather than genus or family level. To be accurate, we used "newly identified virus" in the revised manuscript and clarified the definition of "newly identified virus species" in the Materials and Methods section as following: "Species assignment of these confirmed viral genomic sequences was carried out firstly following the International Committee on Taxonomy of Viruses (ICTV) species demarcation criteria of each genus¹⁵. ICTV species demarcation criteria of each genus are listed in Supplementary Table 5. If ICTV lacks clear criteria in the genera, we used a threshold of amino acid identity of 90% for the viral RdRp (RNA viruses) or conserved replication-associated proteins (DNA viruses)^{54,55}. If a detected virus sequence was below the 90% identity threshold, it was designated a newly identified virus species." (Page 23, Line 516-523)

11. Line 119-121, page 6: "Phylogenetic analysis based on RdRp showed that all the five novel species formed a new cluster distinct from all previously known species in genus Hunnivirus (Fig. 2a)." Same concern as above point, do these sequences provide evidence of a novel species? The criteria for Picornavirus species are quite well defined and the authors need to describe these sequences correctly without hype.

[Response]: We totally agree with the reviewer's concern, and recognize the challenges to virus species classification by meta-transcriptomics as described by Cobbin *et al.* [reference 49 in the revised manuscript]. As mentioned in the response to comment #10, we have provided the definition of "newly identified virus species". Currently, ICTV does not have species demarcation criteria for genus *Hunnivirus*, though ICTV provides general species demarcation criteria for *Picornaviridae* family. Therefore, we used the classification criteria suggested by previous report [reference 52,53 in the revised manuscript], and modified the sentence as: "Phylogenetic analysis based on polymerase protein showed that the identified virus sequences in this study formed a new cluster distinct from previously known virus species in genus *Hunnivirus* (Fig. 2a, Extended Data Fig. 1a)" (Page 6, Line 126-128)

12. Line 521-525, page 23, Data Availability. "Sequences assembled in this study were deposited in the National Omics Data Encyclopedia (NODE) (<http://www.biosino.org/node>) (OEZ008244) and are publicly available as of the date of publication. Accession numbers are listed in Extended Data Table 3." In Extended Data Table 3. the authors say the data are in a repository. However, no accession numbers are provided. The assembled genomes should be deposited in GenBank and accession numbers should be provided instead of saying "TBD".

[Response]: Following the reviewer's suggestion, we have submitted all the sequences to GenBank and provided the accession number in the Supplementary Table 1 and 2 of the revised manuscript. Please note that we are still waiting for the accession numbers of five viral sequences, and all the sequences will be released immediately after NCBI processing. In addition, we have provided all assembled sequences in the supplementary data for manuscript review. The Data Availability statement has been modified as: "Pangolin mitochondrial sequences and viral sequences assembled in this study were deposited in the GenBank, and accession numbers are listed in Supplementary Table 1 and 2." (Page 26, Line 590-592).

13. Line 305-307, page 14: "The first limitation of the study is pooling different tissues of pangolins for metagenomic sequencing, from which the distribution of viruses in different organs cannot be observed." The danger of pooling is that virus sequence chimeras can be generated if the individual animals were infected with different but closely related viruses. This should be considered and noted in the analysis that the apparent diversity may be an artefact of pooling.

[Response]: We appreciate this valuable comment. As the response to comment #1, only the tissues from the same pangolin were pooled in one sequencing library. We have revised the sentence as: "The first limitation of the study is pooling different tissues of a single pangolin into one sequencing library, from which the distribution of viruses in different organs or tissues cannot be observed." (Page 15, Line 338-340)

14. Line 299-300, page 13: "In addition, we found pangolin orthoreovirus contain recombination sequence fragments from bat strains." It should be commented that the odd sequence could be an artefact

of assembly from multiple related viruses. Without information of the pooling, it is difficult to assess this validity of these sequences.

[Response]: We appreciated the valuable comment. Following the reviewer's suggestion, detailed pooling information of each library was listed in Supplementary Table 1. We pooled the tissues from each pangolin into a single sample for meta-transcriptome sequencing. We further conducted RT-PCR followed by Sanger sequencing to fill the gaps, and the sequences obtained from RT-PCR were consistent with meta-transcriptome reads in the overlap regions. We revised the results as following: "Interestingly, we observed the signals of reassortment between the pangolin orthoreovirus strain and other strains from masked palm civet (*Paguma larvata*)³² and pig in China³³, bat (*Plecotus auritus*) in Germany³¹, and tree shrew (*Tupaia belangeri*) in China (GenBank accession No. MG451071.1) (Fig. 3d), suggesting cross-species transmission of orthoreoviruses among multiple animal hosts. Phylogenetic analysis revealed discrepancies in the clustering of the seven segments between *Pangolin orthoreovirus* and the strains from other hosts. *Pangolin orthoreovirus* clustered with the bat strain in L1, L2, and S4 segments, with the civet and pig strains in M3 and S2 segments, and with the tree shrew strain in M2 and S3 segments (Fig 3e)." (Page 10, Line 211-224)

15. The use of novel throughout the manuscript is misleading.

For example, Extended data table 1 "novel/known: column,

All of these viruses are known, what are the criteria for calling a hunivirus or pestivirus novel?

Line 36-38, page 3: "Seventeen novel viruses in Hunnivirus, Pestivirus and Copiparvovirus genera formed well-supported monophyletic clusters, which might be pangolin-specific."

Again, no criteria for calling a sequence (not a virus) novel.

[Response]: As mentioned above, we understand and agree with the reviewer's concern. We have combined related information into Supplementary Table 2 of the revised manuscript, and removed the column of "novel/known". The definition of "newly identified virus species" has been described in the response to comment #10. As suggested by the reviewer, we managed to avoid using "novel" throughout the revised manuscript.

Also line 285-286, page 13: "First, we identified 17 pangolin-specific novel viruses in Hunnivirus, Pestivirus and Copiparvovirus genera,"

The statement that these might be pangolin specific is also misleading. Such a result is highly dependent on the rest of the world sampling and simply finding a monophyletic cluster in a sparsely sampled virus family does not make the virus host specific. This statement should be revised carefully.

[Response]: We agree that “pangolin specific” branch might not be host specific due to limited sampling. To avoid the issue, we used “pangolin-associated” in the revised manuscript.

16. Supplementary Table 1. It is not clear what the values represent, perhaps amino acid length? This should be specified in a table legend. "Range of similarity variation compared to NC_023176.1" ◇ The method of calculating this value should be indicated.

[Response]: Following the reviewer’s suggestion, we have provided more details in the table footnote of revised Supplementary Table 3. The table footnote has been added as: “The table presents the amino acid length of each protein for the listed virus sequences. The bottom row of the table shows the amino acid similarity between pangolin pestiviruses and the closest neighbour (GenBank Accession No. NC_023176.1). Sequence similarity was calculated using the p-distance module of MEGA 7.”

The GenBank accession number for each sequence (including the new sequences reported in the manuscript) should be provided.

[Response]: Following the suggestion of the reviewer, we have provided the GenBank accession numbers of each sequence in phylogenetic trees files (newick format) and associated sequence alignments (fasta format) available at <https://doi.org/10.6084/m9.figshare.19499030>. This statement has been added in the Data availability section. (Page 26, Line 593-594)

17. Figure 1. The trees are difficult to read, the authors should decide on the phylogenetic feature they are hoping to illustrate with each tree and prepare an appropriate subset of the reference sequences. For example, the picornavirus tree is just a swarm of lines with a blue box in the middle. Was it necessary to include so many Caliciviridae and Picornavirales sequences in one tree?

[Response]: We agree that there were too many sequences in the *Picornavirales-Caliciviridae* clade. We have reduced the number of sequences in the *Caliciviridae* and *Picornavirales* while keep the overall backbone structure in the revised Fig 1b.

18. The use of the term epizootic is misleading. The word in standard usage means broadly infecting a species and it is difficult to conclude from sampling of a small set of pangolins in one area that any of the virus sequences indicate an epizootic infection. This term should be removed as it is misleading.

[Response]: Thanks for the advice. We have removed the word “epizootic” throughout the revised manuscript.

Responses to reviewer 2:

This manuscript describes a study of the virome of Malayan pangolins confiscated from the illegal wildlife trade in China. The authors identify a large number of viruses, some that form new clades within known viral families and thus appear to be pangolin-specific, and others that cluster with non-pangolin viruses and appear to be recently transmitted to pangolins from other species. The topic is timely, given suspicions that pangolins were somehow involved in the early transmission of SARS-CoV-2 (I believe these hypotheses have largely fallen by the wayside).

The analyses are very well done. In particular, the “forest” of phylogenetic trees presented speaks to the skills of this research group in phylogenetic analyses. The figures are presented clearly, and the figures and other data support the phylogenetic inferences within the paper.

My main critique is not with the data or analysis, but rather with the organization and interpretations. These pangolins were confiscated from the illegal wildlife trade, and the conditions of their capture and transport are unknown. The authors mention that some of the viruses they found could have been transmitted to pangolins from other species, but this idea is not discussed very much. This point should, in my opinion, be central to the manuscript.

As it stands, the manuscript is a “litany of viruses” paper (i.e. “Look at all the viruses we found!”). However, the study would be more interesting and more important if it were framed as follows: Smuggled Malaysian pangolins harbor some viruses that look like “pangolin viruses” and other viruses that look like they were transmitted from other species, including humans. In other words, there are two types of viruses here: pangolin viruses, and non-pangolin viruses. This distinction is buried in the text. It should be a main focus, from the abstract to the discussion. This is important because it is widely believed is that mixing together of animals like pangolins with other animals like bats and civets (both identified as viral hosts in the paper) is a major risk factor for disease emergence. This is the “wet market” hypothesis – that the conditions of wildlife transport and sale create conditions that favor cross-species transmission of viruses. These authors have found compelling evidence in support of that hypothesis, and this evidence is, in my opinion, their most important finding.

[Response]: We appreciate the reviewer’s constructive suggestion for the overall structure of the manuscript. We did reframe the manuscript to emphasize the two types of viruses in pangolins and discussed the “wet market” hypothesis in the revised manuscript as “The presence of diverse viruses in smuggled pangolins suggest that the conditions of wildlife transportation and sale might facilitate cross-species transmission of these viruses, and will possibly lead to viral emergence in captured wildlife and in wet markets.” (Page 15, Line 334-337)

Finally, all the pangolins were one species. The authors imply in several places that they have studied the “pangolin virome,” but it is really the “Malayan pangolin virome.”

[Response]: Thanks for the suggestion. We have modified “pangolin virome” to “Malayan pangolin virome” throughout the revised manuscript.

In light of these points, the authors might consider a title such as: “Virome of illegally trafficked Malayan pangolins reveals novel pangolin associated viral lineages and infection with known pathogens of humans.”

[Response]: We agree with the idea, and revised the title as “Virome of illegally trafficked Malayan pangolins reveals pangolin-associated viral lineages and infection with known pathogens of humans”

Specific comments

Page 4, line 24: Should “gamey meat” be changed to “wild game?” I think this may be an issue with translation.

[Response]: In response to reviewer’s suggestion, we revised “gamey meat” to “game meat” in the revised manuscript.

Page 5, lines 4-6. It is unclear what is meant by “lack of species classification of pangolins.” Are the authors referring to difficulty with morphological identification of pangolin species, or to lack of DNA sequences from pangolins of known species in public databases? Also, the mitochondrial contigs should be deposited into a public database. I suspect there was variability among pangolins in contig length and mitogenome coverage, so species assignment criteria (e.g. E-values) should be provided.

[Response]: As suggested, we have clarified the description as: “Because only archived tissue samples were available, it was impossible for us to identify the pangolin species through traditional morphological classification. We then identified pangolin species based on mitochondrial contigs in the meta-transcriptome data, and found all 161 pangolins were *Manis javanica*.” (Page 5, Line 85-88).

Species assignment criteria has been provided in the manuscript as following: “For each sample, we compared assembled contigs against a database of all the available pangolin mitochondrial sequences using blastn with an E-value $<1e-5$. Pangolin species were inferred based on the top hits.” (Page 25, Line 577-578). The length of mitochondrial sequences obtained from each pangolin varied from 889 bp to 16504 bp, all of which have been deposited into GenBank and the accession numbers have been listed in Supplementary Table 1. Please note that all the sequences will be released immediately after NCBI processing. In addition, we have provided all assembled sequences in the supplementary data for manuscript review.

Page 7, lines 1-5. It is very interesting that the L-protein has been lost from these viruses is an important and interesting finding. If space permits, consider mentioning this in the Abstract, because it increases confidence in the ability to detect pangolin-adapted viruses.

[Response]: Following the reviewer’s suggestion, we have added one sentence to the abstract as: “ We found the L-protein has been lost from all viruses in genus *Hunnivirus* identified from pangolins.” (Page 3, Line 40-41)

Page 7, lines 11-13. Many pestiviruses appear to be benign. The authors should not over-emphasize the pathogenicity of pestiviruses as a genus.

[Response]: In response to the reviewer’s comment, we have revised the sentence to “Sequences of nine virus species were newly identified in the genus *Pestivirus* of family *Flaviviridae*, which is known to cause asymptomatic infection or produce a range of clinical conditions such as acute diarrhea, acute hemorrhagic syndrome, acute fatal disease, and a wasting disease in pigs and ruminants^{21,22}.” (Page 7, Line 144-147)

Page 8, line 20. The novel Orthopneumovirus is interesting, but it could be an unknown human variant or a virus or from another species that pangolins have encountered during smuggling. It would be better to call this virus something other than “pangolin orthopneumovirus” in case it is identified as actually being a virus of another species.

[Response]: Thanks for the suggestion. We have revised the virus name as “*Orthopneumovirus BIME 1*”. (Page 9, Line 185)

Page 9, lines 11-17. The pangolin-civet connection screams of wildlife trafficking and “wet markets.” Pangolins and civets, along with bats, are the most often suspected reservoirs or amplifying hosts of viruses when they are mixed together. The authors should be sure to emphasize this fact (see suggestion above for reframing the paper).

[Response]: We appreciated the valuable comment. Following the reviewer’s suggestion, we first conducted RT-PCR followed by Sanger sequencing to fill the gaps, and revised the results as following: “Interestingly, we observed the signals of reassortment between the pangolin orthoreovirus strain and other strains from masked palm civet (*Paguma larvata*)³² and pig in China³³, bat (*Plecotus auritus*) in Germany³¹, and tree shrew (*Tupaia belangeri*) in China (GenBank accession No. MG451071.1) (Fig. 3d), suggesting cross-species transmission of orthoreoviruses among multiple animal hosts. A phylogenetic analysis revealed discrepancies in the clustering of the seven segments between *Pangolin orthoreovirus* and the strains from other hosts. *Pangolin orthoreovirus* clustered with the bat strain in L1, L2, and S4 segments, with the civet and pig strains in M3 and S2 segments, and with the tree shrew strain in M2 and S3 segments (Fig 3e).” (Page 10, Line 211-220)

We have also expanded our discussion as following: “The presence of diverse viruses in smuggled pangolins suggest that the conditions of wildlife transportation and sale might facilitate cross-species transmission of these viruses, and will possibly lead to viral emergence in captured wildlife and in wet markets.” (Page 15, Line 333-336)

Page 10, lines 14-15. Ticks can be highly host specific, so the risk of cross-species transmission might not be very high.

[Response]: We agree with the reviewer’s opinion. We have revised the sentence to: “The presence of tick-associated virus sequences in pangolins reveals risks for transmission among vertebrates through tick vectors.” (Page 13, Line 281-283)

Page 12, lines 6-7. It is unclear what is meant here by “sufficient number of mitochondrial variants.”

[Response]: In response to the comment, we have clarified the description as: “Because all the Malayan pangolins were smuggled to China without any information on their original habitats, we conducted phylogeographic analysis based on pangolin mitochondrial variants to infer their population groups. After excluding pangolin samples with poor quality data of mitochondrial variants, 128 Malayan pangolins in this study together with other 11 Malayan pangolins with known geographic origins as well as mitochondrial sequences⁴⁸ were included for the analysis.” (Page 13, Line 285-291)

Page 12, line 25. “Protected” should be used in place of “preserved.”

[Response]: Done as suggested.

Page 13, line 8. The conclusion that “Our findings indicate that pangolins harbor diverse viruses with zoonotic and epizootic risks.” Is an overstatement and not very novel. With the alternative framing suggested above (pangolins harbor a mix of pangolin viruses and non-pangolin viruses), the authors could write a more compelling final paragraph. An important point is that the non-pangolin viruses were likely acquired during smuggling, in support of the “pangolin-bat-civet” hypothesis of viral emergence in captured wildlife and in markets.

[Response]: We are grateful for the constructive suggestion, and have revised the discussion paragraph around the two categories of viruses identified in pangolins of this study as following “In this study, we identified a mix of pangolin-associated as well as non-pangolin virus sequences. All the sequences of 16 pangolin-associated virus species belonged to three genera (i.e. *Hunnivirus*, *Pestivirus* and *Copiparvovirus*), with high genetic diversity in each genus. The prevalence of these virus species among pangolins were relatively high. Furthermore, the genomes of viruses in genus *Hunnivirus* identified in pangolins all lost L-protein region, suggesting the virus might have adapted to pangolins. We speculate that pangolins might be natural reservoir hosts of these viruses. The sequences of 12 non-pangolin virus species identified in pangolins were phylogenetically related to those of viruses from various animals, such as civet, bats, rodents, dogs and even humans. The deficient immune system of pangolin makes it more susceptible to viral infections¹³ from other wildlife or humans especially during illegal trade. For instance, a pangolin sample (P251T) simultaneously contained sequences of *Rotavirus A*, Malayan-CoV-HKU4, *Shannbavirus* and *Protoparvovirus*, further supporting the susceptibility of pangolins and possible

27complex exposure to various animals⁵⁰. The presence of diverse viruses in smuggled pangolins suggest that the conditions of wildlife transportation and sale might facilitate cross-species transmission of these viruses, and will possibly lead to viral emergence in captured wildlife and in wet markets.” (Page 14, Line 319-336)

Lines 14-18. From what I know, there has been no documented case of a pangolin virus being transmitted to humans or to cause human epidemics. It would be good to tone this statement down. However, the authors should definitely keep the conclusion that pangolins could infect humans. Hopefully, that will deter poachers (but I doubt it, unfortunately).

[Response]: We are grateful for the suggestion. The paragraph of this sentence has been completely revised to emphasize the cross-species transmission in wildlife transportation and sale. Please see the response to previous comment.

Page 21, lines 11-14. What PCR assays were used to verify the three viruses?

[Response]: In response to the comment of reviewer 1, we performed RT-PCR assays in the available samples of the detected viruses in meta-transcriptome sequencing, and have revised the description as “To further verify the presence of identified virus sequences rather than from contamination, we performed RT-PCR assays in the available samples of the detected viruses in meta-transcriptome sequencing. The specific RT-PCR primers designed according to assembled virus sequences are listed in Extended Data Table 1. The target RT-PCR products were validated by Sanger sequencing (Supplementary Fig. 1).” (Page 24, Line 551-556)

Page 22, line 11. “Determine” is too strong here. Choose a word that reflects the uncertainty inherent in phylogenetic analyses.

[Response]: Thanks for the suggestion. We have used the word “understand” instead of “determine” in the revised manuscript.

Page 23, line 11. It would be good to deposit the assembled viral and pangolin mtDNA sequences in GenBank so that they will appear in BLAST searches at NCBI. The authors relied heavily on NCBI for their analyses, so they should deposit their sequences in NCBI too.

[Response]: As suggested, we have deposited the assembled viral and pangolin mtDNA sequences in GenBank. The accession numbers of these sequences are list in Supplementary Table 1 and 2. Please note that we are still waiting for the accession numbers of five viral sequences, and all the sequences will be released immediately after NCBI processing. In addition, we have provided all the assembled sequences in the supplementary data for manuscript review.

Figure 3. The rotavirus A tree implies that the infected pangolins might have been infected from a common source. Perhaps these 7 pangolins were kept in the same cage and contracted rotavirus from the same person, perhaps also transmitting it to each other. The authors should discuss such possibilities, because these sorts of observations are what one would expect from the horrific conditions in which trafficked wildlife are kept.

[Response]: As suggested by the reviewer, we have discussed the possibility in the revised manuscript:

“Phylogenetic analysis based on RdRp revealed that the sequences of pangolin rotavirus A formed two close clusters (Fig. 3b, Extended Data Fig. 1e). Within each cluster, the sequences were nearly identical, suggesting that the viruses might come from the same pangolin cage or the same wet market during the smuggling and trade.” (Page 9, Line 192-195)

Responses to reviewer 3:

Shi and colleagues present work, titled “Characterization of pangolin virome sheds light on risks of emerging zoonotic and epizootic viruses” that uses meta-transcriptome sequencing to begin to characterise the viromes of pangolin. Pangolins were, until recently, not present in the virology literature, but the discovery of SARS-CoV-2-related coronaviruses in Malayan pangolins in China has highlighted the need for further work. Here authors investigate the virome using samples from 161 pangolins smuggled into China. They discovered number virus genomes or genome fragments and report 29 distinctive vertebrate-associated viruses, with 22 novel viruses. Their work included four “human-associated viruses”, including respiratory syncytial virus, a novel orthopneumovirus, rotavirus A and mammalian orthoreovirus, tick-associated and mammal-associated viruses, including coronavirus closely related to HKU4-CoV from bats. There suggest the discovery of diverse viruses means that pangolins should be considered potential reservoirs or intermediate hosts of zoonotic or epizootic pathogens.

Overall, I think the work is valuable, well performed and presented, the arguments clear and the literature well used. Pangolin are, as the authors report, largely unexplored hosts of viruses (or other infectious diseases). The work here is the natural next step, given previous work on coronaviruses and will be a useful addition to the field. The species sampled and diversity of viruses make it an interesting study. I don't use all the tools (especially the bioinformatics), but the methods used seem standard and the data and methods are provided and therefore repeatable. The results provide many hypotheses for future work and the authors are, I believe, suitably cautious with their interpretations. Here are some comments (not in order of importance).

Throughout the manuscript, unless the journal has its own formatting, the International Committee on Viral Taxonomy formatting for viral nomenclature should be followed. Throughout it seems correct for mammals (for example, class and order level taxa names not italicised) but ICVT recommended italicisation for all (e.g. see online at ICVT FAQs: <https://talk.ictvonline.org/information/w/faq/386/how-to-write-virus-species-and-other-taxa-names>).

[Response]: We appreciate the reviewer's valuable comment, and have italicized all the virus names in the revised manuscript.

Title/line 1: Insert: “Malayan” to the pangolin name, as there is only one species (*Manis javanica*) reported.

[Response]: Done as suggested.

Abstract/line 42: “...HKU4-CoV known in bats...”?

[Response]: In response to this inquiry, we have revised the sentence as “Notably, a coronavirus related to HKU4-CoV initially reported in bats was identified.” (Page 3, Line 44-45)

Intro/line 56: I believe the demand for pangolin is largely from Asia, but the authors are right that North America is involved with the trade, based on the literature, but the text is slightly misleading as written. I would rephrase this and use peer-reviewed literature to clarify these relationships, e.g. Heinrich et al, *Global Ecology & Conservation*, 2016 “Where did all the pangolins go? International CITES trade in pangolin species”.

[Response]: In response to reviewer’s suggestion, we have revised the text as: “They are considered the world’s most trafficked wild mammal due to the consumer demand for their meat and scales largely in Asia, and North America is also involved with the trade as recorded in the Convention on International Trade in Endangered Species of Wild Fauna and Flora (CITES) Trade Database⁵.” (Page 4, Line 55-59)

Line 90/Fig 1a: I agree with these statement regarding ‘vertebrate-associated’ viruses, but I would like some clarification around some of the viruses and their host associations. In Fig 1a (and elsewhere) some of the vertebrate host associations are likely, and evidence strong, even for viral families associated with a range of hosts, but for some it’s not clear. For example, in Fig 1a, the rhabdovirus sequences are classified as vertebrate, but the Rhabdoviridae family has a wide host association and not all are vertebrate associated and it’s not clear from the tree in Fig 1b which rhabdoviruses the new sequences are most close to. Can the authors review and revise/confirm the putative viral-host relationships?

31[Response]: We are very grateful for the reviewer's suggestion, and have revised the host range of the viral families in Fig 1a as well as rewritten the legend accordingly as following: "For the viral families with multiple host categories, the host category of each viral read was inferred based on the host of assigned genus given by blastx with E-value <1e-05." (Page 30, Fig 1 legend)

Though the viral reads of family *Rhabdoviridae* were detected in a pangolin sample as shown in Fig. 1a, there was no assembled contigs containing RdRp domain in family *Rhabdoviridae*, which cannot be shown in Fig. 1b. We have added a note in the legend of Fig. 1 as: "Families without assembled contigs containing RdRp domain were not displayed in Fig. 1b, though the viral reads were detected in Fig. 1a" (Page 31, Fig 1 legend)

Lines 162/311: A general point, with respect to viruses associated with humans, how sure can the authors be that there was not simply contamination of samples with human viruses/tissues, rather than true infection? For example, the Human respiratory syncytial virus subtype A (RSV-A) data is convincing (including because of the pangolin orthopneumovirus BIME 1) but where the authors write: "...we cannot trace their original infection sites of the viruses", this can include a comment on contamination of samples, or more details in the methods about how this might have been mitigated? Given the source (trafficked animals) and how these might have been collected it seems contamination rather than infection (viral replication) must at least be acknowledged.

[Response]: We appreciated the valuable comments, and have provided detailed description for controlling possible contamination during library preparation and sequencing in the revised manuscript. In the Materials and Methods section, we added "To identify and eliminate possible sequencing or reagent contaminants, we included a sterile water as well as reagent mix as controls⁵¹. The controls were sequenced in the same chip with the libraries of pangolin samples." (Page 22, Line 498-501)

In addition, methods to control cross-talk contaminant was described as following: "We compared the presence of viral sequences and their abundance across sequencing libraries. In case the identical or near-identical viral sequences were observed in multiple libraries, we used the following steps to rule out possible contamination. We first estimated the read ratio between the highest abundance library and other lower abundance libraries in the same chip. If the ratio was below the index-hopping rate of the sequencing platforms, the reads in low abundance library were considered as cross-contamination during library preparation and excluded from further analysis⁵¹. To eliminate possible index-hopping, we chose the highest percent of 0.1% as the threshold^{59,60} in the study. To further verify the presence of identified

virus sequences rather than from contamination, we performed RT-PCR assays in the available samples of the detected viruses in meta-transcriptome sequencing. The specific RT-PCR primers designed according to assembled virus sequences are listed in Extended Data Table 1. The target RT-PCR products were validated by Sanger sequencing (Supplementary Fig. 1). For *Mammalian orthoreovirus*, RT-PCR was also performed to fill gaps in L1 segment which contained RdRp.” (Page 24, Line 544-559).

Furthermore, we confirmed the presence of RSV A in the two samples using RT-PCR followed by Sanger sequencing. We have added the RT-PCR results and biological plausibility in the revised manuscript as following: “We confirmed the presence of RSV-A in the two samples using RT-PCR followed by Sanger sequencing (Extended Data Fig. 4 and Supplementary Fig. 1).” (Page 9, Line 179-183)

Lines 184-186: I agree (I think), but perhaps revise this sentence to: “These findings suggest that rotavirus A might have transmitted long enough among pangolin for viral evolution to lead to new genotypes, and pangolins might”

[Response]: We appreciate the reviewer’s suggestion, and have revised this sentence to: “These findings suggest that *Rotavirus A* might have associated with pangolin for an extensive period of time, alternatively, it could be due to lack of sampling that covering the diversity gap between pangolin rotaviruses and those identified from other mammalian host” (Page 9, Line 201-204)

Line 193: This might be worth discussing very briefly, because recombinant corona-orthoreoviruses have been observed in bats (<https://www.mdpi.com/1999-4915/12/5/539/htm>)

[Response]: We appreciated the valuable comment. In the revised manuscript, we conducted RT-PCR followed by Sanger sequencing to fill the gaps, and revised the results as following: “Interestingly, we observed the signals of reassortment between the pangolin orthoreovirus strain and other strains from masked palm civet (*Paguma larvata*)³² and pig in China³³, bat (*Plecotus auritus*) in Germany³¹, and tree shrew (*Tupaia belangeri*) in China (GenBank accession No. MG451071.1) (Fig. 3d), suggesting cross-species transmission of orthoreoviruses among multiple animal hosts. A phylogenetic analysis revealed discrepancies in the clustering of the seven segments between *Pangolin orthoreovirus* and the strains

from other hosts. *Pangolin orthoreovirus* clustered with the bat strain in L1, L2, and S4 segments, with the civet and pig strains in M3 and S2 segments, and with the tree shrew strain in M2 and S3 segments (Fig 3e). Reassortment has been frequently observed in segmented virus, and is considered as a primary mechanism for interspecies transmission and the emergence of novel strains³⁴. A cross-family recombinant from orthoreovirus and coronavirus has been reported in bats³⁵, suggesting that distinct recombination events occur in orthoreovirus and might increase the potential for spillover.” (Page 10, Line 211-224)

Line 195: “Pangolin were easily...” implies they were infected here. Maybe revise this to “...were previously reported to be heavily infested with ticks ...” or “...may be heavily infested with ...”

[Response]: We appreciate the suggestion, and have revised the sentence to: “Malayan pangolins were previously reported to be heavily infested with ticks...” (Page 12, line 262-264)

Line 281: I agree, this is a good number of samples, but honestly, pangolin confiscations have included up to 120 tonnes of whole of pangolin, there are ample opportunities for more sample collection. Though hopefully this will cease, perhaps the authors could make a recommendation that sample collection is included in animal seizures?

[Response]: Thanks for the valuable suggestion. We have add the sentence in the revised manuscript: “To better evaluate the viral composition and potential risks to the public health, we propose that sample collection should be included in animal seizures.” (Page 14, line 317-319)

Line 470/471: This seems fair, but not that ICVT has classification processes, which the authors used for the rotavirus discussion. Did the authors look at all viruses using ICVT thresholds?

[Response]: Thanks for the comment. We have included the ICTV classification criteria and revised the classification process in the Materials and Methods section as following: “Species assignment of these confirmed viral genomic sequences was carried out firstly following the International Committee on Taxonomy of Viruses (ICTV) species demarcation criteria of each genus¹⁵. ICTV species demarcation criteria of each genus are listed in Supplementary Table 5. If ICTV lacks clear criteria in the genera, we used a threshold of amino acid identity of 90% for the viral RdRp (RNA viruses) or conserved replication-associated proteins (DNA viruses)^{54,55}. If a detected virus sequence was below the 90% identity threshold, it was designated a newly identified virus species.” (Page 23, Line 516-523)

Line 501-502: did the authors look at ambiguously aligned regions? (And if so, consider trimming, e.g. TrimAI, and realignment)?

[Response]: We did look at the ambiguously aligned regions in the sequence alignment. In response to the reviewer’s inquiry, we have added a sentence in the Materials and Methods section: “The ambiguously aligned regions were trimmed using TrimAI and manually edited⁶².” (Page 25, Line 566-567)

Line 504: my guess is it won’t change the overall findings, given the tree topologies and branch supports, but did the authors try another approach, e.g. MrBayes to confirm the topologies, especially for key relationships?

[Response]: Thanks for the valuable suggestion. We have further conducted phylogenetic analysis using MrBayes software. The topologies of Bayesian trees are similar to those of maximum likelihood trees. We have added MrBayes analysis method in the Materials and Methods as “To further confirm the topology of the phylogenetic tree of each virus genus, phylogenetic analyses were performed using MrBayes v3.2.7⁶⁴ (10 million generations) with the same models with the maximum likelihood method. The MrBayes trees were provided in Extended Data Fig. 1.”. We have indicated the corresponding Bayesian trees simultaneously with the maximum likelihood trees in the results section (e.g. Fig. 2a, Extended Data Figure 3a) of revised manuscript.

Fig 6: can you increase the circle and point size?

[Response]: Done as suggested.

Decision Letter, first revision:

Dear Dr. Cao,

Thank you for submitting your revised manuscript "Virome of illegally trafficked Malayan pangolins reveals pangolin-associated viral lineages and infection with known pathogens of humans" (NMICROBIOL-21123116A). It has now been seen by the original referees and their comments are below. The reviewers find that the paper has improved in revision, and therefore we'll be happy in principle to publish it in *Nature Microbiology*, pending minor revisions to satisfy the referees' final requests and to comply with our editorial and formatting guidelines.

Thank you again for your interest in *Nature Microbiology*. Please do not hesitate to contact me if you have any questions.

Sincerely,

{redacted}

Reviewer #1 (Remarks to the Author):

Comments on the revised manuscript "Virome of illegally trafficked Malayan pangolins reveals pangolin-associated viral lineages and infection with known pathogens of humans " (19185 _revision)

The revised manuscript has substantially improved. The authors have addressed most of my previous concerns, but there are still a few minor points to be corrected:

1. Line 937. " The deficient immune system of pangolin makes it more susceptible to viral infections¹³ from other wildlife or humans especially during illegal trade."

36I wonder where the evidence is that pangolins have a "deficient immune system". The reference 13 is a general review and I see no data documenting any immune deficiency in these animals. This statement either be supported with references or be removed.

2. Line 1224. "To identify and eliminate possible sequencing or reagent contaminants, we included a sterile water as well as reagent mix as controls⁵¹. The controls were sequenced in the same chip with the libraries of pangolin samples. "

Rebuttal: "The controls were sequenced in the same chip with the libraries of pangolin samples."
(Page 22, Line 498-501)"

The data from sequencing water controls should be shown as well as indicating other samples sequenced or handled on the same run. I don't find the water control data in the revised manuscript.

3. Supplementary Table 1. Basic information of each sample used for meta-transcriptome analyses

Footnotes explaining "Accession No.(mitochondrion)" and "Length of mitochondrion" should be provided. Also, the table rows should be numbered.

Reviewer #2 (Remarks to the Author):

The authors have done an admirable job responding to reviewer comments. I particularly like the new section on pangolin phylogeography. I also appreciate the revised figures and the re-written sections of the Discussion, which is now more nuanced. Overall, there are still some awkward and unclear sentences likely due to translation issues (although the authors' English is much better than my Chinese!), but these can be corrected during the editorial process. A few specific and very minor suggestions follow.

Line 37. "Newly identified" would be a better phrase than "first identified."

Lines 57-57. I understand that the authors are responding to another reviewer here, but I wonder if it wouldn't be better simply to delete the phrase "largely in Asia, and North America is also involved with the trade as recorded in the Convention on International Trade in Endangered Species of Wild Fauna and Flora (CITES) Trade Database." I'm sure there are examples of trafficked pangolins in Europe or the Middle East, for example, so it's hard to write a phrase like this without committing an error of omission. A few citations about pangolin trafficking could guide the interested reader to regional statistics.

Lines 76-77. Consider changing "To investigate the prevalence of the emerging coronaviruses in the unique wildlife" to "To investigate the pangolin virome more comprehensively." This might be better because the paper is not only about coronaviruses.

37Line 87. Change "then" to "therefore."

Line 111. Change "distinctive" to "distinct."

Lines 286-297. This is a very nice addition to the manuscript. Some of the text is a bit unclear, however. I suggest that modifications be conducted with the help of the journal editorial staff.

Reviewer #3 (Remarks to the Author):

Dear authors,
I am sorry for being so slow to get to this review and holding this up. Thank you for addressing my comments thoroughly.

Decision Letter, final checks:

Dear Dr. Cao,

Thank you for your patience as we've prepared the guidelines for final submission of your Nature Microbiology manuscript, "Virome of illegally trafficked Malayan pangolins reveals pangolin-associated viral lineages and infection with known pathogens of humans" (NMICROBIOL-21123116A). I've edited your paper, because Dr Troppens is moving to a permanent role on Nature Communications.

Please carefully follow the step-by-step instructions provided in the attached file, and add a response in each row of the table to indicate the changes that you have made.

Please also check and comment on any additional marked-up edits we have proposed within the text, and USE THAT FILE as the basis for your revisions. Ensuring that each point is addressed will help to ensure that your revised manuscript can be swiftly handed over to our production team.

38In recognition of the time and expertise our reviewers provide to Nature Microbiology's editorial process, we would like to formally acknowledge their contribution to the external peer review of your manuscript entitled "Virome of illegally trafficked Malayan pangolins reveals pangolin-associated viral lineages and infection with known pathogens of humans". For those reviewers who give their assent, we will be publishing their names alongside the published article.

Nature Microbiology offers a Transparent Peer Review option for new original research manuscripts submitted after December 1st, 2019. As part of this initiative, we encourage our authors to support increased transparency into the peer review process by agreeing to have the reviewer comments, author rebuttal letters, and editorial decision letters published as a Supplementary item. When you submit your final files please clearly state in your cover letter whether or not you would like to participate in this initiative. Please note that failure to state your preference will result in delays in accepting your manuscript for publication.

Cover suggestions

As you prepare your final files we encourage you to consider whether you have any images or illustrations that may be appropriate for use on the cover of Nature Microbiology.

Nature Microbiology has now transitioned to a unified Rights Collection system which will allow our Author Services team to quickly and easily collect the rights and permissions required to publish your work. Approximately 10 days after your paper is formally accepted, you will receive an email in providing you with a link to complete the grant of rights. If your paper is eligible for Open Access, our Author Services team will also be in touch regarding any additional information that may be required to arrange payment for your article.

Please note that Nature Microbiology is a Transformative Journal (TJ). Authors may publish

39their research with us through the traditional subscription access route or make their paper immediately open access through payment of an article-processing charge (APC). Authors will not be required to make a final decision about access to their article until it has been accepted. [Find out more about Transformative Journals](https://www.springernature.com/gp/open-research/transformative-journals)

Authors may need to take specific actions to achieve [compliance with funder and institutional open access mandates](https://www.springernature.com/gp/open-research/funding/policy-compliance-faqs). If your research is supported by a funder that requires immediate open access (e.g. according to [Plan S principles](https://www.springernature.com/gp/open-research/plan-s-compliance)) then you should select the gold OA route, and we will direct you to the compliant route where possible. For authors selecting the subscription publication route, the journal's standard licensing terms will need to be accepted, including [self-archiving policies](https://www.nature.com/nature-portfolio/editorial-policies/self-archiving-and-license-to-publish). Those licensing terms will supersede any other terms that the author or any third party may assert apply to any version of the manuscript.

Please use the following link for uploading these materials:
{redacted}

Best regards,

{redacted}

Reviewer #1:

Remarks to the Author:

Comments on the revised manuscript "Virome of illegally trafficked Malayan pangolins reveals pangolin-associated viral lineages and infection with known pathogens of humans" (19185 _revision)

The revised manuscript has substantially improved. The authors have addressed most of my previous concerns, but there are still a few minor points to be corrected:

401. Line 937. "The deficient immune system of pangolin makes it more susceptible to viral infections¹³ from other wildlife or humans especially during illegal trade."

I wonder where the evidence is that pangolins have a "deficient immune system". The reference 13 is a general review and I see no data documenting any immune deficiency in these animals. This statement either be supported with references or be removed.

2. Line 1224. "To identify and eliminate possible sequencing or reagent contaminants, we included a sterile water as well as reagent mix as controls⁵¹. The controls were sequenced in the same chip with the libraries of pangolin samples. "

Rebuttal: "The controls were sequenced in the same chip with the libraries of pangolin samples." (Page 22, Line 498-501)"

The data from sequencing water controls should be shown as well as indicating other samples sequenced or handled on the same run. I don't find the water control data in the revised manuscript.

3. Supplementary Table 1. Basic information of each sample used for meta-transcriptome analyses

Footnotes explaining "Accession No.(mitochondrion)" and "Length of mitochondrion" should be provided. Also, the table rows should be numbered.

Reviewer #2:

Remarks to the Author:

The authors have done an admirable job responding to reviewer comments. I particularly like the new section on pangolin phylogeography. I also appreciate the revised figures and the re-written sections of the Discussion, which is now more nuanced. Overall, there are still some awkward and unclear sentences likely due to translation issues (although the authors' English is much better than my Chinese!), but these can be corrected during the editorial process. A few specific and very minor suggestions follow.

Line 37. "Newly identified" would be a better phrase than "first identified."

Lines 57-57. I understand that the authors are responding to another reviewer here, but I wonder if it wouldn't be better simply to delete the phrase "largely in Asia, and North America is also involved with the trade as recorded in the Convention on International Trade in Endangered Species of Wild Fauna and Flora (CITES) Trade Database." I'm sure there are examples of trafficked pangolins in Europe or the Middle East, for example, so it's hard to write a phrase like this without committing an error of omission. A few citations about pangolin trafficking could guide the interested reader to regional statistics.

Lines 76-77. Consider changing "To investigate the prevalence of the emerging coronaviruses in the unique wildlife" to "To investigate the pangolin virome more comprehensively." This might be better

41because the paper is not only about coronaviruses.

Line 87. Change "then" to "therefore."

Line 111. Change "distinctive" to "distinct."

Lines 286-297. This is a very nice addition to the manuscript. Some of the text is a bit unclear, however. I suggest that modifications be conducted with the help of the journal editorial staff.

Reviewer #3:

Remarks to the Author:

Dear authors,

I am sorry for being so slow to get to this review and holding this up. Thank you for addressing my comments thoroughly.

Final Decision Letter:

Dear Professor Cao,

I am pleased to accept your Article "Trafficked Malayan pangolins contain viral pathogens of humans" for publication in Nature Microbiology. Thank you for having chosen to submit your work to us and many congratulations.

42Acceptance of your manuscript is conditional on all authors' agreement with our publication policies (see <https://www.nature.com/nmicrobiol/editorial-policies>). In particular your manuscript must not be published elsewhere and there must be no announcement of the work to any media outlet until the publication date (the day on which it is uploaded onto our website).

Please note that *Nature Microbiology* is a Transformative Journal (TJ). Authors may publish their research with us through the traditional subscription access route or make their paper immediately open access through payment of an article-processing charge (APC). Authors will not be required to make a final decision about access to their article until it has been accepted. [Find out more about Transformative Journals](https://www.springernature.com/gp/open-research/transformative-journals)

Authors may need to take specific actions to achieve [compliance with funder and institutional open access mandates](https://www.springernature.com/gp/open-research/funding/policy-compliance-faqs). If your research is supported by a funder that requires immediate open access (e.g. according to [Plan S principles](https://www.springernature.com/gp/open-research/plan-s-compliance)) then you should select the gold OA route, and we will direct you to the compliant route where possible. For authors selecting the subscription publication route, the journal's standard licensing terms will need to be accepted, including [self-archiving policies](https://www.nature.com/nature-portfolio/editorial-policies/self-archiving-and-license-to-publish). Those licensing terms will supersede any other terms that the author or any third party may assert apply to any version of the manuscript.

You can now use a single sign-on for all your accounts, view the status of all your manuscript submissions and reviews, access usage statistics for your published articles and download a record of

43your refereeing activity for the Nature journals.
